# A tidally driven fjord-like strait close to an amphidromic region

Sissal Vágsheyg Erenbjerg[1,2], Jon Albretsen[3], Knud Simonsen[4], Erna Lava Olsen[1], Eigil Kaas[2,5], and Bogi Hansen[6]

[1]Dept. of Fjord Dynamics, Fiskaaling A/S, við Áir 11, FO-430 Hvalvík, Faroe Islands
[2]Niels Bohr Institute, Copenhagen University, Juliane Maries Vei 30, DK-2100 Copenhagen, Denmark
[3]Institute of Marine Research, P.O. box 1870 Nordnes, NO-5817, Bergen, Norway
[4]Dept. of Science and Technology, University of the Faroe Islands, J. C. Svabosgøta 14, FO-100 Tórshavn, Faroe Islands
[5]National Center for Climate Research, Danish Meteorological Institute, Lyngbyvej 100, DK-2100 Copenhagen, Denmark
[6]Faroe Marine Research Institute, P.O. 3051, FO-110 Tórshavn, Faroe Islands

*Correspondence to*: Sissal Vágsheyg Erenbjerg (sissal@fiskaaling.fo)

**Abstract.** The strait, "Sundalagið Norður", is the northern part of a narrow body of seawater separating the two largest islands in the Faroe Islands (Faroes). It has shallow sills in both ends and considerably deeper waters in-between. South of the southern end of the strait there is an amphidromic region for the semidiurnal tides so that the tidal range is much lower south of the strait than north of it. The resulting tidal forcing generates periodically varying inflow of seawater across the northern sill, but only a part of that manages to cross the narrow and shallow southern sill. Combined with a large input of freshwater, this gives the strait a fjord-like character. To investigate how this fjord-like character affects the circulation within the strait and its exchanges with outside waters, a pilot project was initiated to simulate the dynamics of the strait with a high-resolution ocean model for a month. The model simulations show clearly the dominance of tidal forcing over freshwater (estuarine) and wind on time scales up to a day. On longer time scales, the simulations indicate systematic variations in the net flows (averaged over a diurnal tidal period) through both the upper and deeper layers. These long-period variations of net flow in the model simulations are forced by sea level differences between both ends of the strait generated by the dominant fortnightly and monthly tidal constituents (Mf, MSf, Mm, MSm). Harmonic analysis of sea level records from two tide gauges located off each end of the strait demonstrates that this behaviour is not a model artefact and it has pronounced effects on the strait. Not only does it induce long-period – mainly fortnightly - variations in the net flow through the strait, but it also generates variations in the estuarine characteristics. According to the model simulations, periods with net southward flow – typically lasting a week - have a strait-like character with net southward flow almost everywhere. Periods with net northward flow, in contrast, have a more fjord-like character with stronger salinity stratification and a southward counter-flow in the deep layer. This also induces a large difference in renewal rate of the deep water between the two periods, which is important to consider for human utilization of the strait, especially the local aquaculture plant. The combination of topographic, freshwater, and tidal characteristics creating these long-period variations is rather unusual and it is not known whether similar systems exist elsewhere, but the long-period variations tend to be masked by the stronger semidiurnal and diurnal variations and may easily be overlooked.

## 1 Introduction

This study presents results from a strait, "Sundalagið Norður" [pronounced: ˈsʊntaˌlɛaːjɪ ˈnoːɹʊɹ], located in the Faroe
Islands (hereafter: Faroes), in the North East Atlantic (Fig. 1a). When compared to better known narrow straits (Gibraltar
Strait, Sunda Strait, Strait of Dover, etc.), the strait treated here is considerably smaller, both in terms of physical extent and
volume transport. A priori, Sundalagið Norður therefore might not seem worthy of much interest, but it does have some
features that distinguish it from a typical shallow strait and make it difficult to put into established classification systems
(e.g., Li et al., 2015):

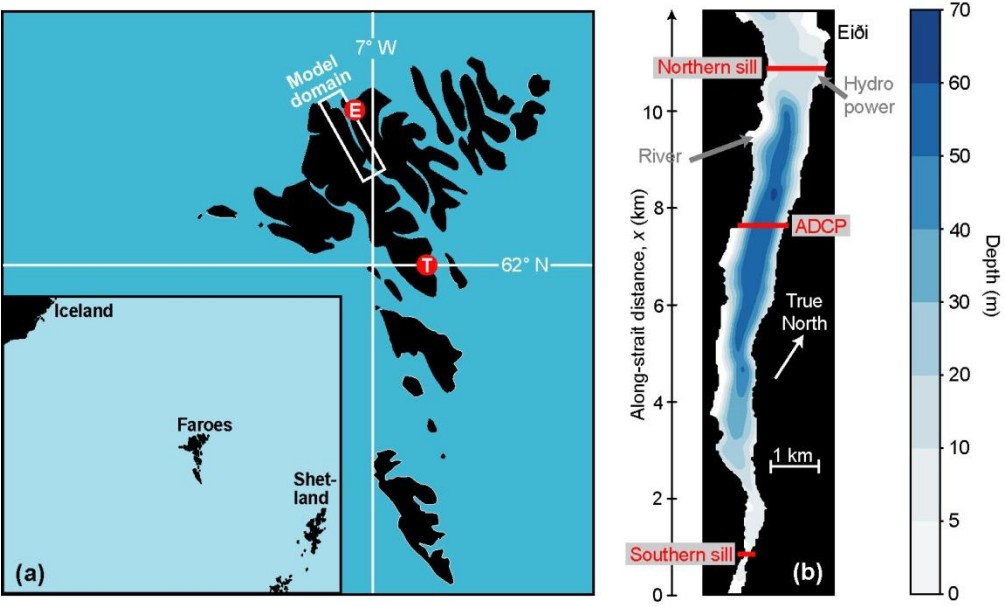

**Figure 1. (a)** The strait studied in this paper is located within the white rectangle, which defines the domain of a high-resolution model
used in the study. The Faroes are situated between Iceland and Shetland (inset map). The red circles show two sites: Tórshavn (T)
[pronounced: ˈtʰɛuːɹ̥ʂ ˌhavn] and Eiði (E) [pronounced: aiˑjɪ] referred to in the text. **(b)** Bottom topography of the strait. Freshwater supply
from a river and a hydro-power plant are shown by grey arrows. Red lines indicate three cross-strait sections discussed in the manuscript.
The vertical x-axis indicates along-strait distance.

Firstly, the strait has sills in both ends that are considerably shallower (sill depths: 4 and 11 m, respectively) than the
central parts (up to more than 60 m). The cross-sectional area at the southern sill (Fig. 1b) is much smaller (670 m$^2$) than the
cross-sectional area at the northern sill (> 12 000 m$^2$). As will be shown, this has the consequence that less than half of the
water entering the strait across the northern sill during the rising tide passes across the southern sill, on average. The
remainder leaves the strait again across the northern sill during the ebbing tide as in a typical tidally driven fjord.

Secondly, the high mountains on both sides of the strait induce high precipitation rates and high runoff into the strait.
On average, the naturally occurring daily runoff into the strait is 0.13 % of the volume between the sills. If the water in the
strait were not continually replenished by saline ocean water, this amount of freshwater would lower the salinity by 0.3 psu
per week, averaged over the total volume of the strait. Since the freshwater will tend to be concentrated in the uppermost

brackish layer, the effect on that layer will be even more and the runoff from a hydropower station adds to this natural freshwater supply. These features indicate that this body of water might behave more like an estuary than a strait, and hydrographic observations in the 1980s revealed that the stratification of the strait was highly variable, but often with a pronounced brackish top layer. This layer, which was typically 10 – 20 m deep, could be less saline than the oceanic water north of the strait by one psu or more. Also, it was observed that the bottom waters of the strait would often become stagnant

during summer with reduced oxygen concentrations near the bottom as is common for Faroese sill fjords (Hansen, 1990). These characteristics were the motivation for using "fjord-like strait" in the title of this manuscript.

Thirdly, there is an amphidromic region on the Faroe shelf that is located close to Tórshavn (Fig. 1a), close to the southern sill of the strait (Hansen, 1978). Most of the attention has been given to the $M_2$-tide (Simonsen, 1992; Simonsen and Niclasen, 2021), but the amphidromic character of the region close to Tórshavn includes the other main semidiurnal

constituents. Also the dominant diurnal constituents are low in this region (Supplementary Fig. S1). With a large difference in tidal amplitude between both ends of the strait, strong tidal currents may be generated and the flow across the southern sill reaches very high speeds according to local fishermen.

Thus, this strait experiences strong tidal forcing as well as estuarine (freshwater) forcing. In addition, the winds may be quite strong and might also affect the flow considerably. A priori, it is not clear, which of these forcing mechanisms

dominate the flow and exchange within the strait, as well as with the surrounding waters. Although small compared to most other straits, Sundalagið Norður may therefore present an interesting case to study from a purely academic point of view. Added to that are questions of a more societal character. Along the coasts on both sides of the strait, a number of villages release sewage and other effluents into the water that may affect the natural biota in various ways. The strait is also hosting a fish farm and there is a potential for negative effects both on and from this activity, as well as interactions with other Faroese

fish farming sites.

On this background, the main aim of this study was to understand how the different forcing mechanisms (freshwater, tidal, wind) combine to generate the physical conditions in the strait. More specifically, the aims are to clarify: 1) How the forcing mechanisms control the flow through the strait and its exchanges with waters outside the strait. 2) How the stabilizing effect of freshwater input competes with the de-stabilizing effects of tidal and wind forcing to affect the

stratification in the strait. 3) What controls the renewal (flushing) rate of the waters in different parts of the strait.

To answer these questions, a numerical model is essential. Previous modelling efforts of the region have mainly been based on barotropic two-dimensional models (Simonsen and Niclasen, 2021; Kragesteen et al., 2018). Rasmussen et al. (2014) and Erenbjerg et al. (2020) have also reported results from full three-dimensional model simulations of the Faroe shelf, but both those models were too coarse to resolve the conditions in the strait.

We have therefore implemented a high-resolution model (32 m × 32 m horizontal, 35 vertical layers) that is one-way triply-nested (Supplementary Fig. S2) within a ROMS (Regional Ocean Model System) model covering a larger region (Lien et al., 2013). Due to limited computing resources, the high-resolution model was only run for 29 days. This period is too short to simulate the generation and decay of bottom layer stagnation. Instead, a period in February-March 2013 was chosen,

mainly because current velocity observations from two ADCP (Acoustic Doppler Current Profiler) deployments in the strait
were available. The model was run for this period with realistic atmospheric forcing, but detailed runoff data were not available, so a constant freshwater supply was prescribed.

## 2 Material and methods

We define our strait to be the region between the two sills. The sills are defined by minimum cross-sectional (east-west) area and located at the red lines in Fig. 1b. The total volume between the sills is $2.31 \cdot 10^8$ m$^3$ with a surface area of $8.75 \cdot 10^6$ m$^2$.

### 2.1 Observations

During the simulation period, two upward-looking Acoustic Doppler Current Profilers (ADCPs) were deployed on the bottom of the strait (Fig. 1b). Details are documented in Larsen et al. (2014a, b). From 2012 to 2018 there were a high number of CTD (Conductivity Temperature Depth) observations in the strait, most of them documented in Simonsen et al. (2018). Unfortunately, no hydrographic observations were made during the simulation period. Sea level measurements are
available from tide gauges at two sites Eiði and Tórshavn (Fig. 1a) from the Faroese Office of Public Works (Landsverk). These were sampled every ten minutes from 2009 to 2014.

### 2.2 The model

We have applied a model setup based on the open-source ROMS model (http://myroms.org, Shchepetkin and McWilliams, 2005; Haidvogel et al., 2008). This is a state-of-the-art three-dimensional hydrostatic, free-surface, primitive equation
solving ocean model. ROMS applies generalized terrain following s-coordinates in the vertical and regular horizontal grids. This setup applies 32 m × 32 m resolution in the horizontal and 35 vertical layers. The triply-nested setup (Supplementary Fig. S2) is forced along the four open boundaries by SVIM (4 km × 4 km horizontal resolution, Lien et al., 2013). The first nesting has a resolution of 800 m in the horizontal and was run for the whole of 2013 (Erenbjerg et al., 2020). The second nesting contains a 160 m horizontally resolved grid and is run for five months in 2013. This second nesting is used as forcing
for the ultra-high-resolution (32 m) setup used in our current study. The 32 m model has 682 grid point along the strait and 187 points in the perpendicular direction and covers a wider area, but we will focus on the region of the strait (Supplementary Fig. S3).

Atmospheric forcing is provided by the Weather Research and Forecasting (WRF) model on the surface. The WRF-model is set up with a configuration that has a resolution of 9-3-1 km in the horizontal and the area with 1 km × 1 km
resolution covers the entire Faroe Islands. More detail on configuration can be found in Myksvoll et al. (2012).

No time series of runoff were available for the simulation period. Therefore, freshwater input to the strait from runoff is assumed to be constant in time and based on two reports: Erenbjerg (2020) and Davidsen et al. (1994) as well as data from the local energy supplier. The two main freshwater sources are a hydropower plant on the eastern coast with annually

averaged runoff 5.5 m³ s⁻¹ and a river on the western coast with annually averaged runoff 2.0 m³ s⁻¹ (Fig. 1b). These values were used as input to set up ROMS.

The computationally demanding high-resolution (32 m) model was run from the 11th of February until the 12th of March 2013. The starting date was two weeks after the start of the 160 m model, which started four weeks after its parent (800 m) model. Since the 160 m model has many mesh points within the strait, the starting conditions for the 32 m model should be approximately realistic so that the spin-up period ought to be relatively short. This is verified by inspection of the temporal evolution of parameters (especially kinetic energy) during the start period of the 32 m model. To avoid any remaining spin-up effects, results from the first day have nevertheless been omitted. Thus, the model output comprises 672 hourly values (from 12 February 01:00 to 12 March 00:00) of velocity and hydrographic parameters for each grid cell as well as sea level. This period will in the following be referred to as the simulation period.

## 2.3 Model validation

### 2.3.1 Validation of tidal characteristics

The tidal forcing is an essential component of the dynamics of the strait and checking whether the tides are adequately simulated is an important part of model validation. To check this, the characteristics (Amplitudes and Greenwich phase-lags) of the main tidal sea level constituents were determined from tide gauge observations at two locations, Tórshavn and Eiði (Fig. 1a), by harmonic analysis and compared with characteristics from simulated sea level close to these locations. One of the locations (Tórshavn) is outside the domain of the high-resolution (32 m) model. Therefore, the comparison was made with the 800 m parent model, which was run for a longer period and therefore also more suitable for harmonic analysis than the 32 m model. The U_TIDE software package, which is the Python adaption of the T_TIDE Matlab version (Pawlowicz et al., 2002), was used for the harmonic analysis. For better comparability, the analysis of the observed sea level data was made for the same period as for the simulated data.

For the observation-model comparison, the dominant three semidiurnal, two diurnal, two fortnightly, and two monthly constituents were selected (Table 1). At the location Eiði, there was good correspondence between observations and model for the two strongest constituents, $M_2$ and $S_2$, in strength (amplitude) as well as timing (Greenwich phase-lags). The correspondence for the other constituents at Eiði was not impressive. At Tórshavn, none of the constituents was very accurately simulated.

A priori, this might be used to conclude that the model does not simulate the tidal sea level variations in the area well. For the tidal forcing of the strait, the important aspect is, however, not the ability of the model to simulate individual constituents, but rather the strength and timing of the sea level difference between both ends of the strait. This difference will to a large extent be determined by the strength and timing of $M_2$ and $S_2$ at Eiði since the amplitudes of these constituents are much larger than the other amplitudes in Table 1. For our purposes, the tidal forcing of the strait, thus, is fairly accurately

simulated by the model, although the difference in Greenwich phase-lag for $M_2$ at Eiði (19° equivalent to 39 minutes) and other differences imply a timing bias in the model, which ought usually not to exceed one hour.

**Table 1.** Amplitudes and Greenwich phase-lags, as determined by harmonic analysis, for nine tidal constituents (Const.) of sea level variation at two locations (Tórshavn and Eiði, Fig. 1a) as observed at tide gauges (Obs.) and as simulated at grid points close to the locations by the 800 m parent model (Model). The harmonic analysis was for the period 1 January 2013 to 1 October 2013 for both the 155 observed and the simulated sea level. For the semidiurnal and diurnal constituents, the periods are listed in hours (h.). For the long-period constituents, they are listed in days (d.).

| Const. | Period | Tórshavn | | | | Eiði | | | |
|---|---|---|---|---|---|---|---|---|---|
| | | Amplitude | | Greenw. phase-lag | | Amplitude | | Greenw. phase-lag | |
| | | Obs. | Model | Obs. | Model | Obs. | Model | Obs. | Model |
| $M_2$ | 12.42 h. | 9.3 cm | 3.2 cm | 195° | 162° | 57.9 cm | 59.6 cm | 249° | 268° |
| $S_2$ | 12.00 h. | 5.1 cm | 4.0 cm | 214° | 193° | 19.9 cm | 21.1 cm | 283° | 285° |
| $N_2$ | 12.66 h. | 1.8 cm | 0.1 cm | 170° | 277° | 12.2 cm | 4.0 cm | 225° | 10° |
| $K_1$ | 23.93 h. | 4.4 cm | 2.6 cm | 139° | 251° | 9.3 cm | 7.3 cm | 159° | 289° |
| $O_1$ | 25.82 h. | 7.2 cm | 3.9 cm | 52° | 315° | 7.2 cm | 3.5 cm | 12° | 269° |
| Mf | 13.66 d. | 2.7 cm | 2.0 cm | 179° | 213° | 2.0 cm | 1.9 cm | 173° | 215° |
| MSf | 14.78 d. | 3.1 cm | 2.2 cm | 172° | 190° | 2.9 cm | 2.4 cm | 146° | 193° |
| Mm | 27.59 d. | 3.2 cm | 1.2 cm | 186° | 163° | 2.9 cm | 1.2 cm | 184° | 153° |
| MSm | 31.81 d. | 4.0 cm | 2.3 cm | 269° | 249° | 4.1 cm | 2.0 cm | 262° | 247° |

**2.3.2 Comparison of simulated and observed current velocity**

The two ADCPs were located on a transect crossing the strait (Fig. 1b), but high-quality data (Larsen et al., 2014a, b) were only obtained for the deep parts of the velocity profiles (Fig. 2a). When averaged over the simulation period, simulated and 160 observed cross-strait profiles are similar and close to zero (Fig. 2b and 2c). For mooring site BW (Fig. 2b), the average simulated and observed along-strait profiles are also fairly similar for the depths reached by the ADCP. For mooring site BE (Fig. 2c), the discrepancy is larger.

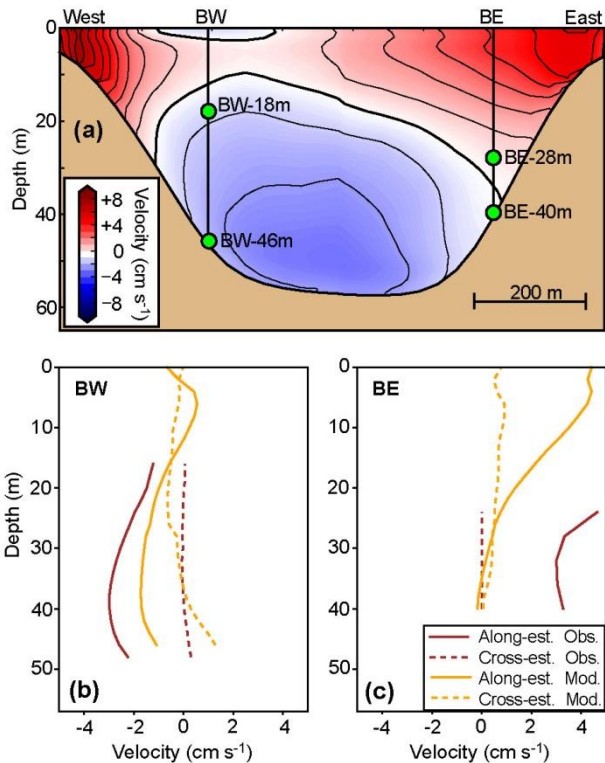

**Figure 2. (a)** The background colours show simulated northward velocity along transect labelled ADCP in Fig. 1b averaged over the simulation period. The two vertical lines labelled BW and BE indicate the locations of two ADCPs moored on the bottom and green circles indicate the uppermost and lowermost depths with high-quality measurement for each of the ADCPs. **(b)** and **(c)** Velocity profiles along the strait (continuous curves, positive towards north) and across it (dashed curves, positive towards east) from the ADCP measurements (dark curves) and the model (light curves) at the two mooring sites averaged over the simulation period.

To evaluate the simulation of temporal velocity variations, Hovmøller diagrams may be used to illustrate the hourly variations of along-strait velocity with depth and time at the two ADCP sites (Supplementary Fig. S4). These diagrams show some similarities between observations and model, but also some differences. A more objective evaluation is presented in Fig. 3, which compares along-strait velocities at two depths for each ADCP site. Although the strength of the tidal forcing is well simulated by the model, as argued above, the timing of flood and ebb may be off by roughly an hour (Table 1). This is not important for understanding the dynamics of the strait, but will affect a direct comparison (correlation) of observed and simulated hourly velocities negatively.

Figure 3, therefore, does not compare hourly velocities directly. Instead, the top panels in the figure compare observed and simulated along-strait velocities averaged over consecutive 25-hour periods while the bottom panels compare the standard deviations within the same 25-hour periods. The length of this period is roughly twice the period of the $M_2$ constituent and intermediate between the $K_1$ and $O_1$ constituents. The top panels of the figure, thus, compare the long-term (longer than daily) variations of along-strait velocity averaged over a diurnal tidal period, while the bottom panels compare

the magnitudes of the variations within a diurnal tidal period, which include the neap-spring variations in the strength of the tidal current.

The observed and simulated velocities in Fig. 3 are certainly not identical, but they are generally of the same magnitudes. All of the correlation coefficients are also positive and six out of eight are significantly higher than zero at the 95 % level (p < 0.05). Although the observations do not validate the model in detail, they do exhibit similarity to the simulations and there is no indication that the model generates unrealistic velocities. Since the ADCP data did not reach into the upper layers with strong tidal currents, they do not provide a strong test of the simulations.

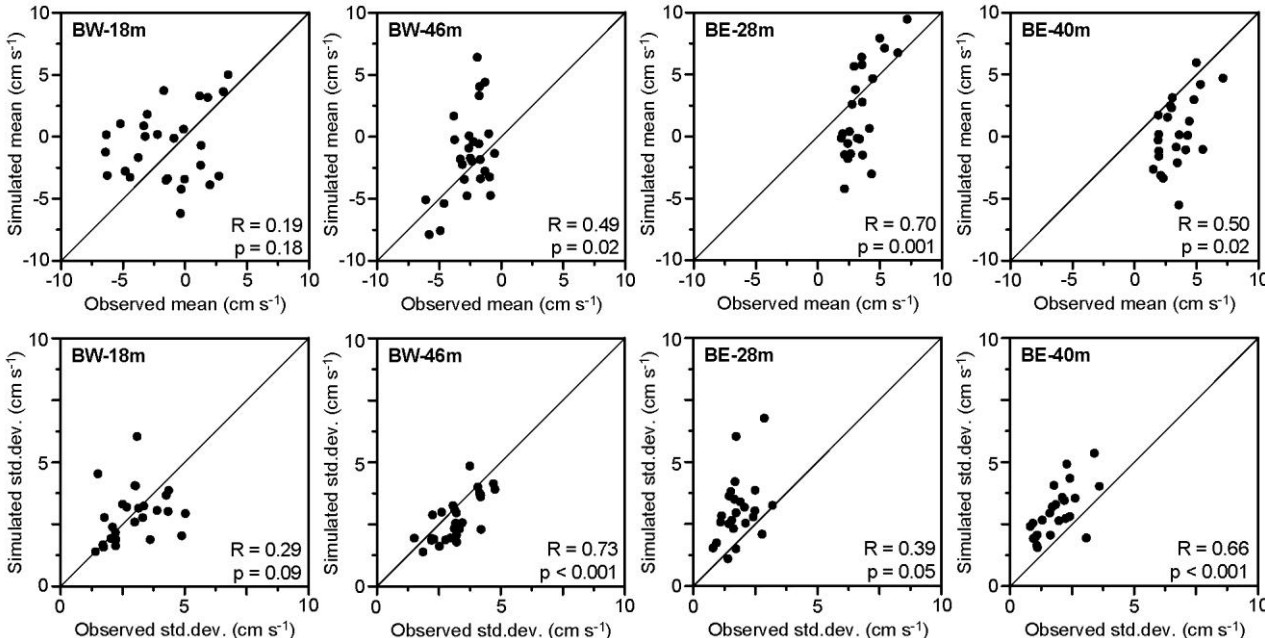

**Figure 3.** Comparison between observation and model for averages (top panels) and standard deviations (bottom panels) of along-strait velocity for consecutive 25-hour periods at two measurement depths for each of the ADCP sites (Fig. 2a) during the simulation period. Correlation coefficients (R) and their statistical significances (p) are listed in the lower right corner of each plot. Diagonal lines indicate equality between model and observation. Here and elsewhere in the manuscript, the statistical significance of correlation coefficients has been corrected for serial correlation by the "modified Chelton" method recommended by Pyper and Peterman (1998).

## 2.3.3 Comparison of simulated and observed salinity

No hydrographic observations were made during the simulation period. Instead, the data from all the CTD observations in the strait 2012 – 2018 were collected (Simonsen et al., 2018). In the shallow regions on either side of the strait, detailed bottom topography and proximity to a river outlet may affect the salinity disproportionately. We therefore considered only CTD casts with bottom depth at least 50 m. To exclude situations with a stagnant bottom layer, only observations from winter (November – April) were used (Supplementary Fig. S5).

At a first glance, the correspondence between observed and simulated salinity is not impressive and might indicate too strong mixing in the model. The model was, however, run with nearly constant freshwater supply. Therefore, it does not simulate periods with excessive runoff that are frequent in the Faroese winter and likely to have caused the observed CTD profiles in the figure with very low salinities in the top 10-20 m layer.

## 3 Results

### 3.1 Hourly variations

To structure the presentation, Fig. 4a defines a few key time series of volume transport and sea level that are sampled hourly. The volume transports are across the northern sill, $q_N$, and the southern sill, $q_S$, respectively, where positive values indicate northward volume transport and negative values indicate southward transport. They are calculated by integration of 210 simulated northward velocity across each of the sill sections taking sea level variations into account. The sea level time series are all cross-strait averaged.

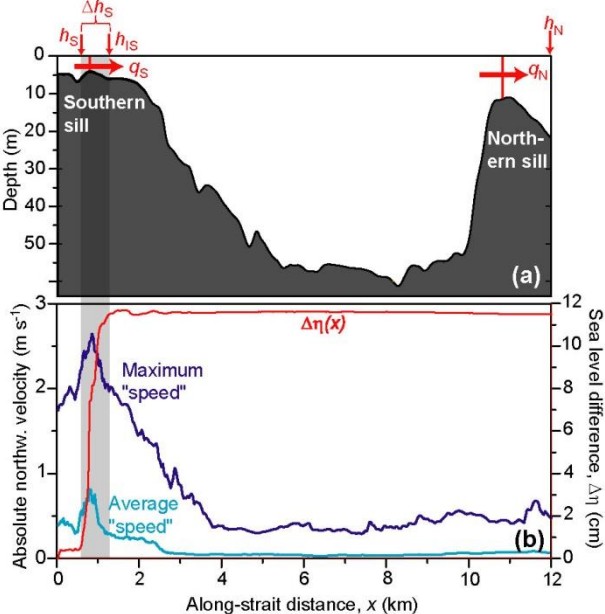

**Figure 4. (a)** Locations for sampling key time series. **(b)** The cyan and dark blue curves show the along-strait variation of maximum (dark blue) and (temporally) averaged "speed", defined as the absolute value of the cross-strait averaged along-strait velocity component. The 215 red curve shows the standard deviation, $\Delta\eta(x)$, of the sea level difference between the southernmost part ($x = 0$) and a given location, $x$.

On hourly time scales, the flow through the strait is clearly dominated by the tides. This is evident in Fig. 5a, which shows the simulated hourly variations of volume transports across both the northern, $q_N$, and the southern, $q_S$, sill during the 4-week simulation in 2013. The semidiurnal variation is clear in the hourly values, as is a fortnightly variation in the amplitude of the transport. The amplitude of the transport across the northern sill is much higher than the amplitude in

transport across the southern sill. From a regression analysis, the highest correlations are found when $q_S$ lags one hour after $q_N$, and the amplitude ratio is then 0.46 (Table 2). As mentioned in the introduction, this implies that most of the water entering the strait from the north will return northwards like in a tidally driven fjord.

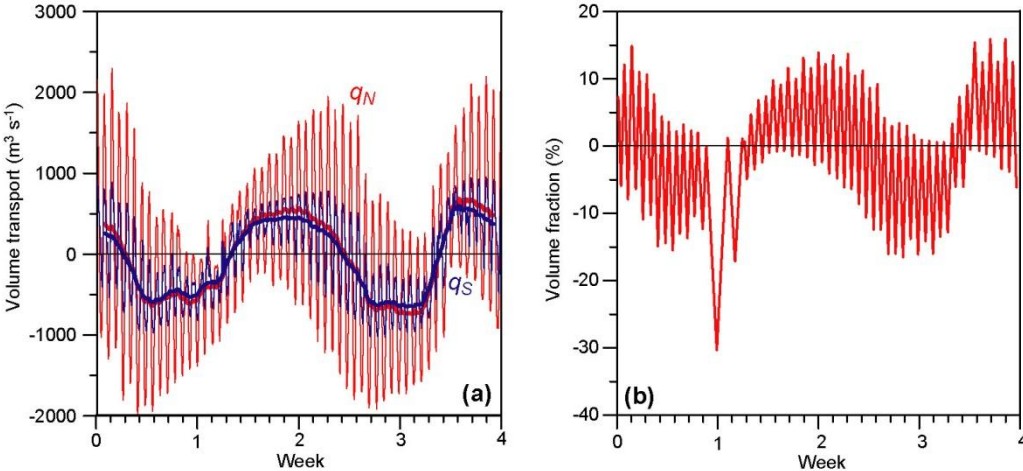

**Figure 5. (a)** Hourly (thin lines) and 25-hour averaged (thick lines) simulated northward volume transport across the northern sill, $q_N$, (red)
and across the southern sill, $q_S$, (blue). **(b)** Fraction of total volume transported into (positive) or out of (negative) the strait across the northern sill during each uni-directional period.

     Most of the time, the volume transport changes sign four times a day, consistent with semidiurnal tidal forcing, although there are a few cases with unidirectional flow lasting more than a day. Adding up all the water flowing into or out of the strait across the northern sill during one of these tidal phases, we find that this typically is around 10 % of the volume,
but occasionally the phases may last longer and transport more water, such as the period by the end of the first week of simulation where an episode of excess southwards flow flushes 30 % of the volume out (Fig. 5b).

     Sea-level variations also follow the tidal cycle. Figure 4a shows three locations, from which hourly sea level time series have been sampled: $h_N$ is sea level north of the northern sill. $h_S$ is sea level south of the southern sill. $h_{IS}$ is sea level just north of the southern sill. In addition, $h_I$ is sea level averaged over all surface grid points between the two sills.

Correlation coefficients (R) between some of these time series are listed in Table 2. Average sea level between the two sills, $h_I$, is very highly correlated with sea level north of the strait, $h_N$, with zero lag (less than one hour) and a regression coefficient ($\alpha$) close to one. Even $h_{IS}$, just north of the southern sill (Fig. 4a), follows $h_N$ almost identically. Thus, sea level within the strait responds more or less instantaneously to the sea level north of the strait. Sea level south of the strait, $h_S$, is also highly correlated with $h_N$, but with a smaller regression coefficient (Table 2).

Hourly values for volume transport across the two sills are fairly well correlated with the difference in sea level between both ends of the strait, $h_N - h_S$, consistent with the idea that this difference drives the flow through the strait. The regression factor, $\alpha_{max}$, is considerably smaller for $q_S$ than for $q_N$, which again is consistent with weaker flow across the southern sill than the northern. In long and shallow estuaries, reflection of the tidal wave may cause behaviour like this (e.g.,

Sohrt et al., 2021). In our case, the wave length of the tidal wave exceeds the length of the strait by more than an order of magnitude. Instead, this difference may be explained by bottom friction over the southern sill, which reduces the flow across the sill. To see this, consider the energy balance in the strait as illustrated in Fig. 4b. The cyan and dark blue curves in that figure demonstrate that the cross-strait averaged speed (and kinetic energy) is much higher over the southern sill than elsewhere in the strait. The red curve shows the along-strait variation of the parameter $\Delta\eta(x)$, which is defined as the typical value (standard deviation) of the sea level difference between the southern end of the strait and the location $x$. This value should be proportional to the typical difference in potential energy between the two locations. As seen in Fig. 4b, this difference remains almost constant throughout most of the strait with most of the change occurring over a relatively short distance over the southern sill with the highest speeds.

**Table 2.** Lagged correlation and regression analysis of relationships between hourly values of various simulated time series where $t$ represents time. $R_0$ is the correlation coefficient for zero lag. $Lag_{max}$ is the lag (in hours) that gives the numerically highest correlation coefficient. $R_{max}$, $\alpha_{max}$, and $\beta_{max}$ are the correlation coefficient and the regression coefficients for that lag.

| Regression equation | $R_0$ | $Lag_{max}$ | $R_{max}$ | $\alpha_{max}$ | | $\beta_{max}$ | |
|---|---|---|---|---|---|---|---|
| $q_S(t+lag)=\alpha \cdot q_N(t)+\beta$: | 0.84 | 1 | 0.87 | 0.46 | | -36 | $m^3\,s^{-1}$ |
| $h_I(t+lag)=\alpha \cdot h_N(t)+\beta$: | >0.99 | 0 | >0.99 | 1.01 | | 0.04 | m |
| $h_{IS}(t+lag)=\alpha \cdot h_N(t)+\beta$: | >0.99 | 0 | >0.99 | 1.01 | | 0.01 | m |
| $h_S(t+lag)=\alpha \cdot h_N(t)+\beta$: | 0.96 | 0 | 0.96 | 0.89 | | 0.00 | m |
| $q_N(t+lag)=\alpha \cdot [h_N(t)-h_S(t)]+\beta$: | -0.88 | 0 | -0.88 | -7606 | $m^2\,s^{-1}$ | 346 | $m^3\,s^{-1}$ |
| $q_S(t+lag)=\alpha \cdot [h_N(t)-h_S(t)]+\beta$: | -0.94 | 0 | -0.94 | -4349 | $m^2\,s^{-1}$ | 165 | $m^3\,s^{-1}$ |

It seems likely that the high kinetic energy over the southern sill is fed by this loss in potential energy. In a simple conceptual model where the speed over the southern sill at any given time, $v_S$, is assumed not to vary spatially on the cross-section, energy conservation may be expressed more rigorously by a modified Bernoulli equation:

$$\tfrac{1}{2} \cdot \rho \cdot v_S^2 = \tfrac{1}{2} \cdot \rho \cdot v_U^2 + g \cdot \rho \cdot \Delta h_U - W_{friction} \qquad (1)$$

where the first term on the right hand side of Eq. (1) is the kinetic energy upstream, which is small and may be ignored. In the next term, $\Delta h_U$ is the sea level difference between the sill and the region upstream, which is either north of or south of the sill, depending on the direction of flow. The last term, $W_{friction}$, is the work done by friction on a water parcel of unit volume. The volume transport is given as $q_S = A \cdot v_S$, where $A$ is the cross-sectional area over the sill, which is assumed to be constant ($= 670\ m^2$). If no energy is lost to friction ($W_{friction} = 0$), this leads to the equation in the upper right hand corner of Fig. 6.

To get a common framework for both flow directions, $\Delta h_U$ may be expressed in terms of the sea level difference across the entire southern strait, $\Delta h_S$ (Supplementary Fig. S6). The locations for defining $\Delta h_S$ are chosen from Fig. 4 as the interval over which $\Delta\eta(x)$ and therefore also the potential energy typically exhibit their main change. This energy-based framework is tested in Fig. 6. Each black point in that figure represents simulated values for $\Delta h_S$ and $q_S$ for one hour, while the red curve

is a least squares fit of the equation in the figure to the simulated values where the parameter $\gamma$ has been varied to minimize

the squared error.

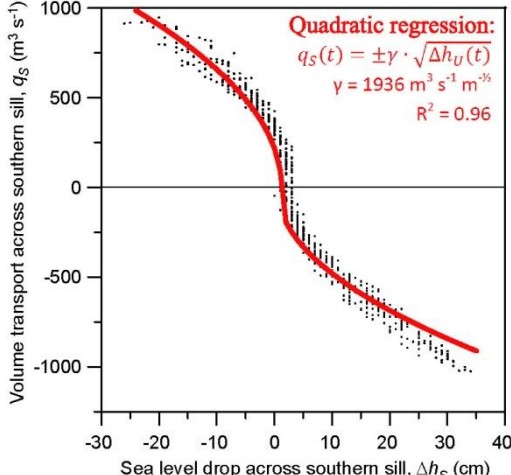

**Figure 6.** Volume transport across the southern sill, $q_S$, plotted against sea level change across the southern sill, $\Delta h_S$. Each black point represents one hour in the model simulation. The red curve represents the result of a least squares fit to the red equation shown in the upper right hand corner (quadratic regression). The quadratic fit (red curve and equation) explains 96 % of the variance of $q_S$.

The fitted expression (red curve) in Fig. 6 appears to represent the simulated values (black points) fairly well, but the

value for $\gamma$ that gives the best fit, 1936 m$^3$ s$^{-1}$ m$^{-\frac{1}{2}}$, is considerably smaller than given by theory with no friction, $\gamma = A \cdot \sqrt{2g}$

= 2969 m$^3$ s$^{-1}$ m$^{-\frac{1}{2}}$, which indicates that friction cannot be ignored. In the model setup, bottom stress was parameterized to

depend on the square of the speed with a drag coefficient of $3 \cdot 10^{-3}$. The work done by bottom friction over a given distance

may therefore also to a good approximation be proportional to $v_S^2$ just as the kinetic energy. In that case, the red equation in

Fig. 6 will remain valid with a lower value for $\gamma$. In this interpretation, only 43 % of the potential energy is converted to

kinetic energy with the rest lost to bottom friction over the southern sill.

    To check whether the high frictional energy loss might be a model artefact, an independent analysis was made on an

idealized conceptual model of the southern sill. In ROMS, as well as in nature, conversion of potential energy into kinetic

energy and loss to friction occur simultaneously as the water approaches and crosses the sill. In the conceptual model

(Supplementary Fig. S7), these two processes occur in separate areas, which better allows distinguishing between them. To

achieve this, the topography of the southern sill in the conceptual model was simplified, but it is still fairly similar to nature

(Fig. 1b and Fig. 4a). With the same drag coefficient ($3 \cdot 10^{-3}$) as ROMS, the conceptual model produces similar values for the

frictional energy loss, supporting the results from ROMS.

### 3.2 Long-period variations

The fortnightly variation of tidal transport amplitude seen in Fig. 5a is a normal phenomenon in Faroese waters (Hansen,

1978), reflecting the variation between neap and spring tides. The net volume transports, averaged over a diurnal tidal period

of 25 hours do, however, also exhibit fairly large long-period (dominant fortnightly) variations (thick lines in Fig. 5a). These variations are further illustrated in Fig. 7, which shows two types of parameters. The symbols that are enclosed in brackets (<>) have been averaged over 25 hours, whereas those that are labelled "std( )" show standard deviations within each 25 hour interval, which should reflect the amplitude of the combined tidal variations from all the semidiurnal and diurnal constituents.

As seen in Fig. 7a, the simulation period includes two full periods with net southward flow and one period with net northward flow, all of them lasting about one week. This figure also has a curve (black) that shows the variation of the standard deviation of $q_N$ over consecutive 25-hour intervals. It shows that the 25-hour average transport does not reach its extremes during spring tide, but rather a few days out of phase (Fig. 7a).

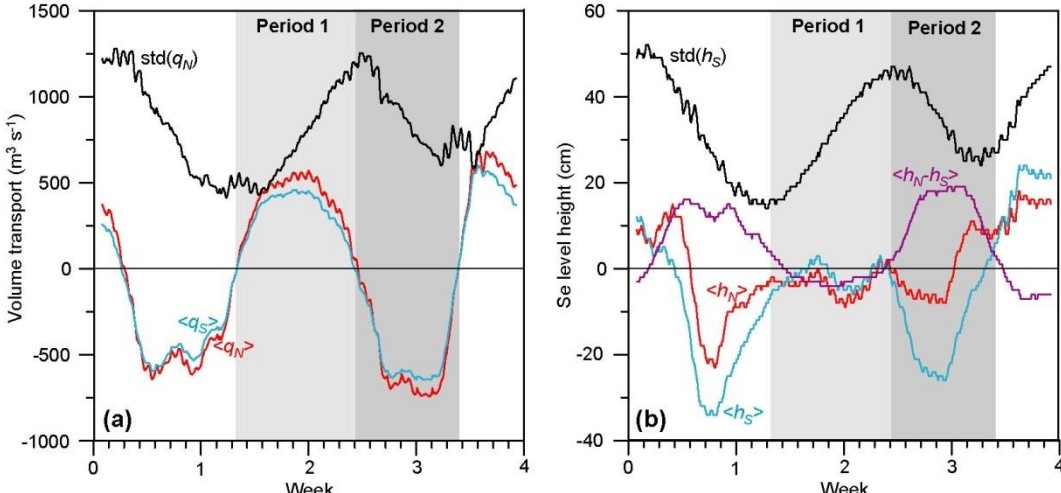

**Figure 7.** Daily (25-hour) averages (<>) and standard deviations (std) within each 25-hour period for **(a)** volume transports and **(b)** sea level heights. The standard deviations ought to be dominated by the strength of the tidal amplitudes and should therefore reflect the variation between spring and neap tides. The shaded areas indicate two periods discussed in the text: Period 1 lasting 188 hours from 21 February 08:00 to 1 March 03:00 and Period 2 lasting 158 hours from 1 March 04:00 to 7 March 17:00.

To help understand the long-period transport variations, Fig. 7b shows 25-hour averages and standard deviations of sea level height. During the periods with average southward transport, average sea level is higher north of the strait. When the average transport is northward, the average sea level is higher south of the strait. Consistent with Table 2, the volume transport through the strait may be seen as forced by the sea level difference between both ends. In this paradigm, the reason for the long-period variations in volume transport is the variation in this sea level difference.

Figure 8 shows the selected two full periods of northward and southward flow, respectively. Period 1 is the first full period with average northward flow, whereas Period 2 is the following period with average southward flow. To illustrate the differences between these two periods (and to the whole-period averages, Supplementary Fig. S8), northward velocity and salinity are averaged over each period and across (east-west) the strait and then plotted against along-strait distance and depth in Fig. 8.

During period 1 (upper panels in Fig. 8), the average flow is northwards through the upper parts of the strait, down to 20−30 m depth except for the water over the northern sill. This indicates that during this period the upper parts of the strait are refilled by the water entering from the south. In the bottom part of the strait, Period 1 has average southward flow. Across the northern sill a thin layer of seawater is entering the sill with a velocity of up to 2 cm s[-1] on average. This denser water manages to descend towards the bottom of the basin before losing its excess density due to mixing and continues southwards at depth.

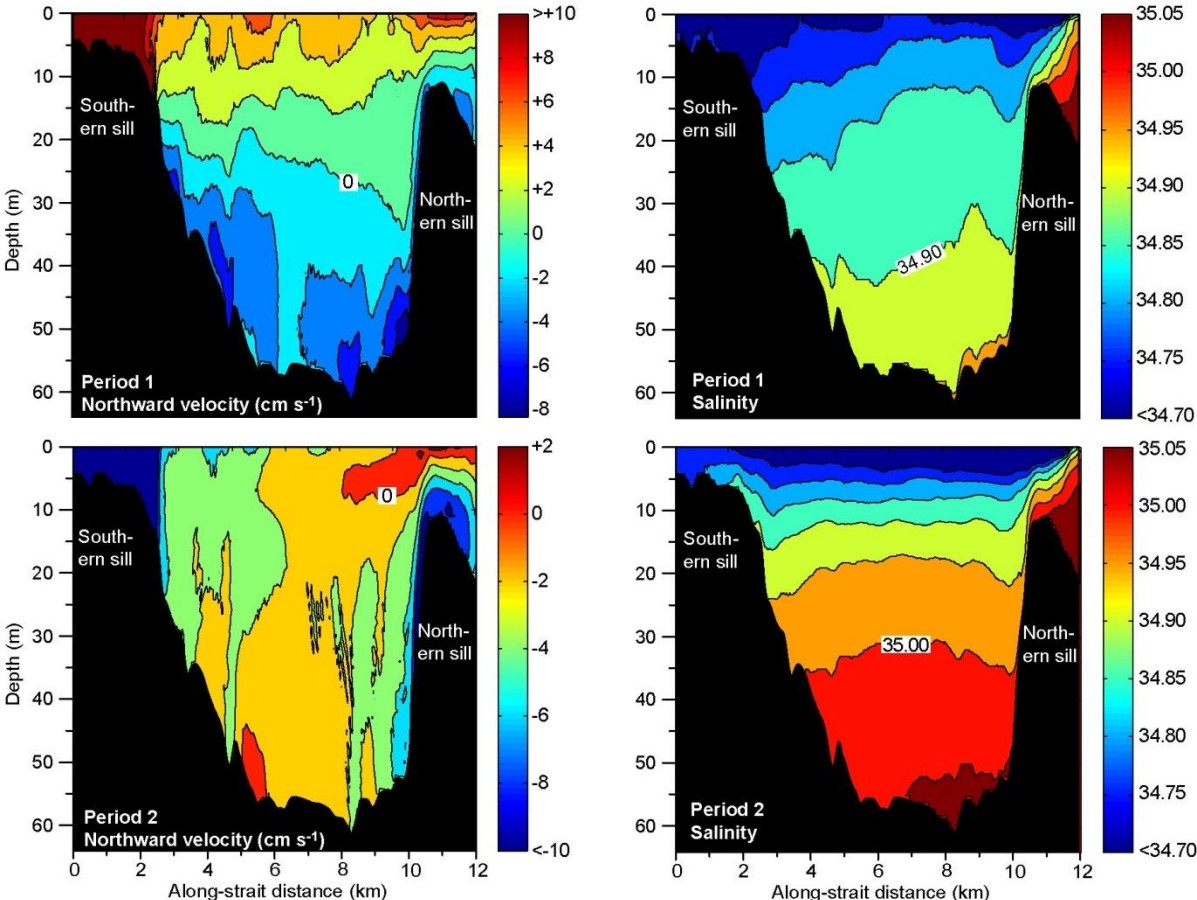

**Figure 8.** Cross-estuary averaged northward velocity and salinity plotted against along-strait distance for the two different periods defined in Fig. 7. For velocity, the most positive (top left panel) or negative (bottom left panel) values are grouped together. Note different velocity scales. For salinity, the lowest values are grouped together. The bottom depth indicated by the black areas is the maximum depth along each section crossing the estuary.

During Period 2 (lower panels in Fig. 8), the average velocity is southwards throughout most of the strait and does not exhibit the fjord-like two-layer circulation of Period 1. The salinity distribution during Period 2 also differs markedly from that of Period 1 with stronger salinity stratification. Since the high-salinity source north of the north of the strait and rate of freshwater supply are almost identical, these differences must be caused by the differences in circulation and mixing.

The difference between the deep flows during the two periods is illustrated in the period-wise averages in Fig. 9, showing volume transport below sill depth of the northern sill (11 m). From this figure it is clear that the volume transport through the deep parts of the strait is much greater during Period 2. This difference is likely one of the main causes of the difference in salinity distributions between the two periods (Fig. 8) and will have a substantial effect on the flushing rate of

the deep waters in the strait as will be discussed in Sect. 4.

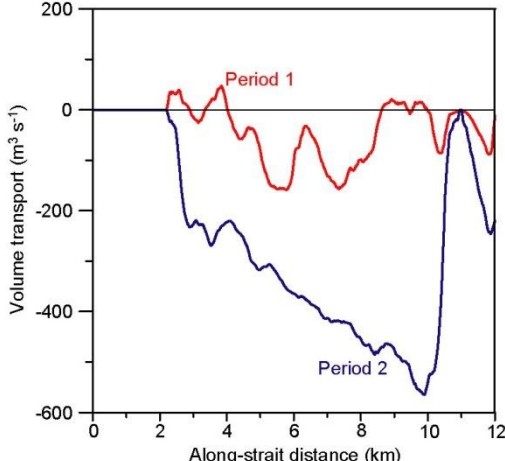

**Figure 9.** Northward volume transport through (east-west) cross-sections below sill depth of the northern sill (from 12 m, downwards) for the two different periods indicated in Fig. 7.

### 3.3 Observational validation of long-period variations

To check that the long-period variations in average transports and sea level differences (Fig. 7) are not purely an artefact generated by the model, we have used the characteristics (amplitudes and Greenwich phase-lags) of tidal constituents derived from tide gauge observations at Eiði and Tórshavn (Fig. 1a) in Table 1 to calculate time series of sea level at the two locations. When averaged over 25 hours, the sea level at both sites is dominated by atmospheric pressure variations (Supplementary Fig. S9), but this effect is strongly reduced when the difference between both locations is considered.

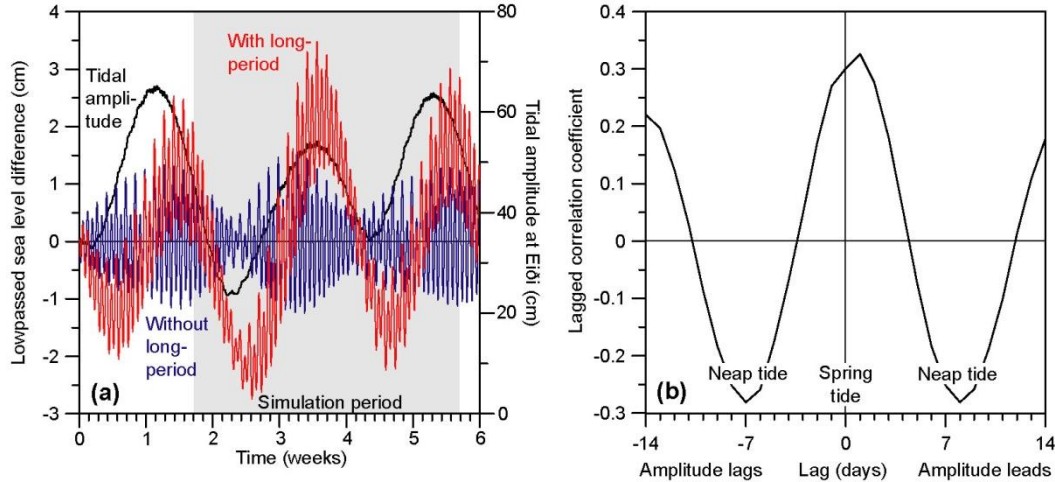


**Figure 10. (a)** Lowpassed (25-hour averaged) sea level difference between Eiði and Tórshavn (Eiði minus Tórshavn) calculated from the observed amplitudes and Greenwich phase-lags in Table 1 for a six-week period starting 1 February 2013. The blue curve (left y-axis) is generated by using only the five semidiurnal and diurnal constituents in the table. The red curve (left y-axis) is based on all the constituents including the four long-period constituents. The black curve (right y-axis) shows the tidal amplitude at Eiði defined as the
standard deviation of sea level for each 25-hour interval. The gray-shaded area indicates the simulation period. **(b)** Lagged correlation coefficient between 25-hour averaged difference in observed sea level between Eiði and Tórshavn and the tidal amplitude in Eiði based on observed sea level (tide gauge measurements) for the 2009-2014 period.

The sea level difference between Eiði and Tórshavn generated in this way therefore ought to be a fair representation of the tidal forcing of the strait and the 25-hour average of this difference (Fig. 10) should reflect the forcing of long-term (longer than a day) variations based on observations. When only semidiurnal and diurnal constituents are used to generate the sea level difference (blue curve in Fig. 10a), the 25-hour averaged difference exhibits long-period variations. They are the residuals of the averaging that result because none of the constituents have periods exactly equal to 25 hours or half of
that. The long-period variation of these residuals is, however, only in the magnitude of their deviation from zero and they change sign several times over a day. Only when the long-period constituents are included (the red curve in Fig. 10a), does the difference exhibit a similar behaviour to Fig. 7b with week-long periods of the same sign.

      Ideally, the curve for $<h_N – h_S>$ in Fig. 7b should be identical to the red curve in Fig. 10a within the shaded area (simulation period), but there are clear differences both in timing and magnitude. This was to be expected from Table 1,
which shows substantial differences between observed (Obs.) and simulated (Model) values for both amplitudes and Greenwich phase-lags for the long-period constituents. In the harmonic analyses of Table 1, the relative uncertainties in amplitude for the long-period constituents are typically 50 % or higher while the uncertainties of the Greenwich phase-lags range between 30° and 90°. The differences between observation and model for the long-period constituents are therefore within the uncertainties of the harmonic analyses.

With so large uncertainties, the question arises whether the signal is real. To address that question, Fig. 10b shows a lagged correlation plot between the observed 25-hour averaged sea level difference between Eiði and Tórshavn and the amplitude of the tidal variation at Eiði based on tide gauge observations from the two locations for the whole 2009-2014 period. For zero lag, the correlation coefficient is positive (R = 0.30). This may perhaps be explained by the residuals from 25-hour averaging of semidiurnal and diurnal constituents, which should be in phase with the amplitude of the tidal variation

at Eiði. The correlation coefficient is slightly higher (R = 0.33) for a positive lag of one day, which is similar to the red curve in Fig. 10a where the highest positive values of low-passed sea level difference lag after the largest tidal amplitude. Most notable are, however, the negative correlation coefficients for lags of ± 7 days, which again is consistent with the red curve in Fig. 10a. All of these correlation coefficients are highly significant (p << 0.001) and demonstrate that the long-period variation illustrated by the red curve in Fig. 10a is not generated by uncertainties in the harmonic analyses.

From this, it seems clear that the long-term variations in Fig. 7 are real and that they are caused by the long-period constituents, mainly the four that are listed in Table 1. Possibly, these variations are enhanced in the model simulations relative to nature, but that is difficult to assess without better observational evidence of the sea level variations, especially for the amphidromic region south of the strait.

### 3.4 Density inversions downstream from the northern sill

From the salinity distributions in the right hand panels of Fig. 8, it seems that the seawater entering the strait across the northern sill is flowing downwards and is then mixed with the water inside the strait just downstream of the sill. To study this in more detail, the density structure was plotted along a track (Fig. 11a). The track was chosen based on the average maximum velocity close to the bottom, beginning on the shallow part of the sill (track number 0) down-slope to 55 m depth (track number 35). The average velocity during the entire simulation period is more than 10 cm s$^{-1}$ southwards in the bottom

layer on the steep slope of the track (Fig. 11d).

     As long as turbulent mixing is weak, the density structure is normally stable with density increasing downwards. Density inversions with density decreasing downwards may therefore be used as a sign of mixing. To utilize this, we define a "density inversion" as the density difference between the third lowest layer and the bottom layer (upper minus deeper) when this value is positive. Under stable conditions, when this value is negative, it is set to zero. The spatial and temporal

variations of density inversions along the selected track are illustrated on the Hovmøller diagram in Fig. 11b.

     If the density inversions are averaged over the track and temporally smoothed, they exhibit systematic variations (red curve in Fig. 11c). Most pronounced is the rather frequent occurrence of inversions in the first part of the simulation from day two to nine. From day nine to fifteen, the density inversions are less frequent for most part of the track except for track points 20-25. These variations show some similarity to the variations in Fig. 7a as shown by the blue curve in Fig. 11c.

Strong southward flow across the northern sill seems to be a necessary – but not sufficient – condition for density inversion. When averaged over time, the density inversions mainly occur on the down-hill slope, and are maximal right before the bottom slope levels off at track number 20, Fig. 11e.

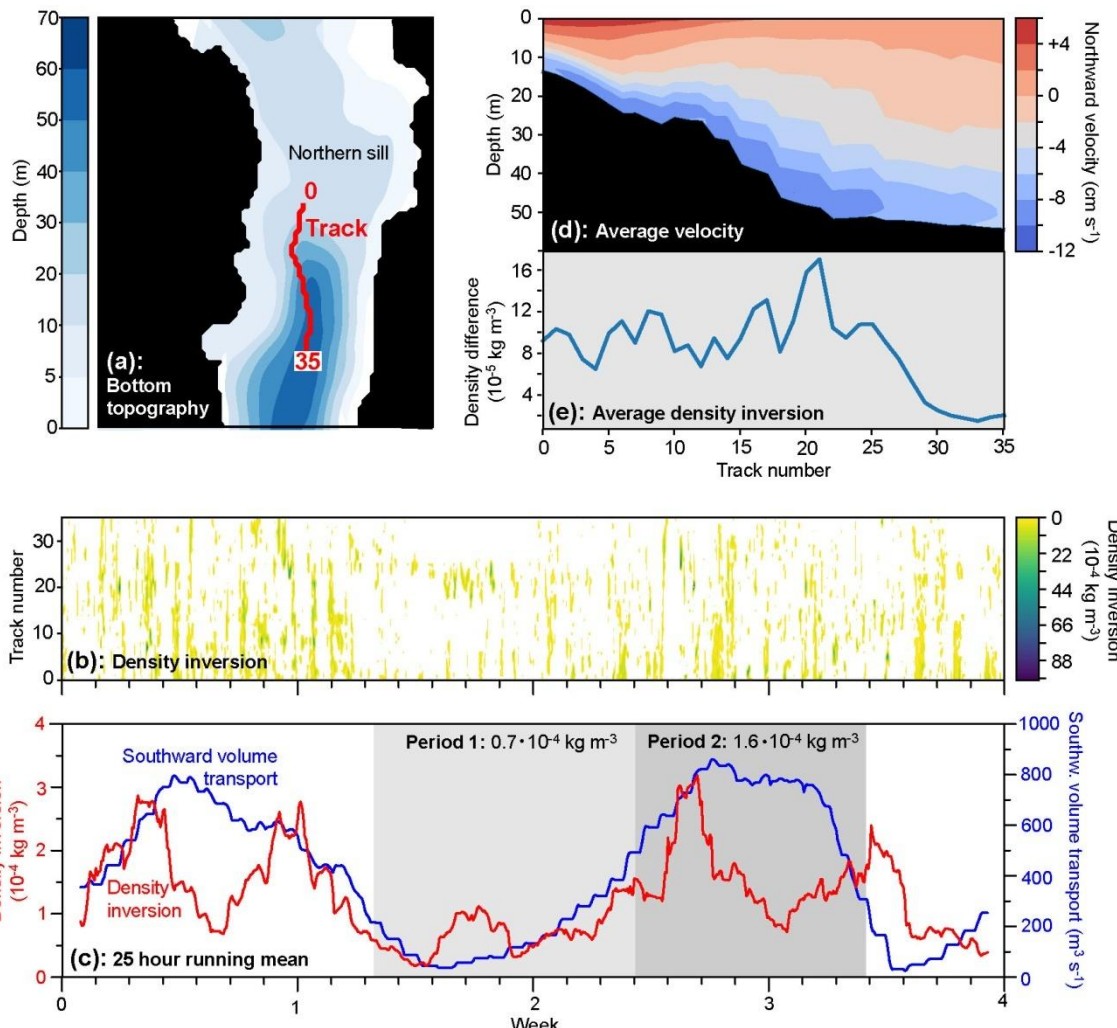

**Figure 11. (a)** Bottom topography around the northern sill. The red line shows a track with numbers ranging from 0 to 35, which are used in the other panels. **(b)** Density inversion, defined as density ($\sigma_\theta$) difference between the deepest layer and two layers above (layer 2 minus layer 0) along the track every hour. Only positive values are shown (negative values white). **(c)** The red curve shows the density inversion in the panel just above averaged over all track numbers and 25-hour running mean. The blue curve is 25-hour running mean of southward-directed volume transport (positive towards the south) across the northern sill (set to zero for hours with northward transport). The shaded areas indicate the two periods defined in Fig. 7 with average density inversion for each period shown. **(d)** Average northward velocity at various depths along the track. **(e)** Density inversion along the track averaged over the whole period.

## 4 Discussion

### 4.1 Model performance

The comparison between tidal constituent characteristics as observed by tide gauges and simulated by the parent (800 m) model verified that the main constituents, which dominate the tidal forcing, were well simulated. The average velocities in the simulation and in observations (Sect. 2.3) did not show identical values, but did agree to a certain extent. For temporal

velocity variations, most of the correlations between model and observations (Fig. 3) were significant at the 95 % (p <0.05) level, which again is encouraging.

For the 800 m parent model, a comprehensive validation against hydrographic observations was performed (Erenbjerg et al., 2020), but no hydrographic (CTD) observations were made during the simulation period of the 32 m model. Instead, salinity profiles from the simulation and from historical CTD data were compared (Supplementary Fig. S5). The correspondence between these two data sets was not impressive and may perhaps indicate that the model has too strong mixing. The model was, however, run with constant runoff rate and can therefore not reproduce the periods with excessive runoff that occur in nature. This aspect of the model setup was unfortunate in terms of model validation, but it has the benefit that it excludes variations in freshwater supply as cause of the long-period variations in salinity distribution (Fig. 8) and estuarine characteristics.

## 4.2 Hourly variations

On short time scales, the flow through the strait is clearly of tidal character with semidiurnal dominance (Fig. 5a) and with sea level differences between both ends as the main driving force. This is supported by the high correlations between volume transports across the sills, $q_N$ and $q_S$, and sea level difference $h_N - h_S$ (Table 2).

As the tidal wave enters the strait from the north, it should propagate southwards as a barotropic wave with sufficient speed to pass through the strait in less than half an hour. This is consistent with the high zero-lag correlations between $h_N$, $h_I$, and $h_{IS}$ (Table 2). The indication of a one-hour lag between $q_N$ and $q_S$ in Table 2 seems strange when all the other time series in the table vary in phase. With data that are sampled every hour, an apparent one-hour difference in lag may, however, be much smaller.

In the model simulations, most of the sea level change occurs over a fairly short distance over the southern sill and this is where the highest speeds are observed (Fig. S4b). Speeds exceeding 2.5 m s$^{-1}$ might seem excessive, but according to local sailors, the speeds over the southern sill may occasionally be considerably higher. From the arguments supporting the quadratic fit in Fig. 6, it appears that roughly half the potential energy in the sea level is lost to friction and from the red curve in Fig. 4b, this loss is over a short distance across the southern sill. Thus, friction over the southern sill probably controls how much water the tides can push through the strait and gives the strait its fjord-like character. The conceptual model presented in Supplementary Fig. S7 supports this conclusion but also emphasizes the importance of choosing an appropriate value for the drag coefficient in the ROMS model. From the literature (Mofjeld et al., 1988; Xu et al., 2017; Wang et al., 2014), the chosen value ($3 \cdot 10^{-3}$) should perhaps have been somewhat lower. This value is, however, quite close to the experimentally derived value, $(2.6 \pm 0.2) \cdot 10^{-3}$, cited by Rippeth et al. (2002) from an area (Menai Strait) similar to the southern sill. This will be an important consideration for any future modelling effort of this strait, but ROMS also has the option of choosing a specific value over the sill, distinct from the value in the rest of the strait.

## 4.3 Long-period variations

As demonstrated in Fig. 5 and Fig. 7, long-period (dominant fortnightly) variations are seen; not only in the tidal amplitude, but also in 25-hour averaged volume transport. Is this an artefact of the processing? On the Faroe shelf, the tides are dominated by semidiurnal and diurnal variations that vary in amplitude over a fortnightly period, mainly as an interference between the $M_2$ tide (period 12.42 hours) and the $S_2$ tide (period 12 hours). For these variations, a 25-hour mean will average out close to zero. There will be a residual, but it should not exceed the maximum tidal amplitude divided by 25. Also, this residual should vary in phase with the strength of the tidal amplitude.

The simulated variations in 25-hour averaged volume transport are much larger than this residual. At times, the 25-hour averaged transport equals the standard deviation of the transport during the same 25 hours and the average transport is not in phase with the amplitude (Fig. 7a). In Sect. 4.2, we argued that the hourly variations in volume transport were forced by sea level differences between the northern and southern ends of the strait. From Fig. 7b, it appears that the same mechanism may be invoked for the 25-hour averaged transport. Periods with average southward transport have average sea levels higher north of the strait than south of it and vice versa.

With this interpretation, the problem is transferred to explaining why there are long-period variations in the 25-hour averaged sea level difference between both ends of the strait. A priori, this might be an artefact generated by the model, but Fig. 10a demonstrates that the long-period variations are real and caused by the dominant fortnightly and monthly tidal constituents (Mf, MSf, Mm, MSm). The similarity between Fig. 7b and Fig. 10a is not perfect. The observed signal in sea level difference (Fig. 10a) is smaller than indicated by the model (Fig. 7b), but this may partly be because Tórshavn is rather far south of the southern end of the strait (Fig. 1a) and in an amphidromic region with large spatial changes.

According to the 800 m parent model simulations, the amplitudes of all the dominant long-period constituents exhibit considerable spatial variations over the Faroe shelf and surrounding areas (Supplementary Fig. S11). To some extent, these variations may be artificial and caused by atmospheric pressure variations that may contaminate the harmonic analysis of long-period constituents for short time series. The lagged-correlation analysis of observed sea level difference between Eiði and Tórshavn (Fig. 10b) does, however, verify the signal independently of harmonic analysis. The positive zero-lag (spring tide) correlation in Fig. 10b could conceivably be caused by residuals from the semidiurnal and diurnal constituents, but the highly significant negative correlations at lags of $\pm$ 7 days (neap tide) are hard to explain without involving the long-period constituents. We therefore conclude that the effects of the long-period tides on the strait are real, although possibly enhanced in the model simulations.

Since southward flow across the southern sill occurs during flood, it has been suggested that the cross-sectional area over the sill – and therefore also volume transport – should be higher during southward than northward flow, which would lead to a net southward volume transport varying between 50 and 175 $m^3$ $s^{-1}$ in phase with the strength of the tidal amplitude (VandKvalitetsInstitutet, 1983). To check whether this is supported by the model simulations, volume transport across the southern sill was calculated from the simulated velocities and sea levels with and without varying sea level. On average, the

difference was 14 m$^3$ s$^{-1}$. Thus, the effect is real, but much smaller than suggested and swamped by the long-period variations.

## 4.4 Exchange rates/ flushing rates

For a water body that is affected by human activity, one of the most important parameters is the flushing rate, i.e., how fast the waters (and dissolved contaminants or planktonic organisms) are flushed out of it. An often used measure of this is the
"flushing time", defined as the volume of the water body (or parts of it) divided by the volume transport into or out of it. Combining CTD observations from our strait with estimated freshwater supply, Hansen (1990) estimated a typical flushing time of 5 days for this strait, but noted the high uncertainty of this value.

From the present model results, there are several ways of obtaining alternative estimates. One way is to use Fig. 5b, which implies that between 5 and 10 % of the volume is typically flushed in and out over the northern sill every 12 hours. If
one assumes no mixing between in-flowing and out-flowing waters, this method gives an average flushing time for the strait as a whole of around one week, ranging between less than four days and more than 11 days over the simulation period (Supplementary Fig. S10).

Alternatively, one could use the average salinity distribution (Supplementary Fig. S8) to estimate the average total freshwater content in the strait and combine that with the (almost constant) freshwater supply to calculate a flushing time.
This method is, however, very sensitive to the choice of salinity for the pure seawater that enters the strait across either of the sills and also assumes stationary conditions.

An extra challenge when trying to estimate a typical flushing time is the long-period variation in the net flow through the strait (Sect. 4.3). Considering the two periods defined in Fig. 7, Figure 12 illustrates the two different exchange regimes that the strait regularly shifts between.

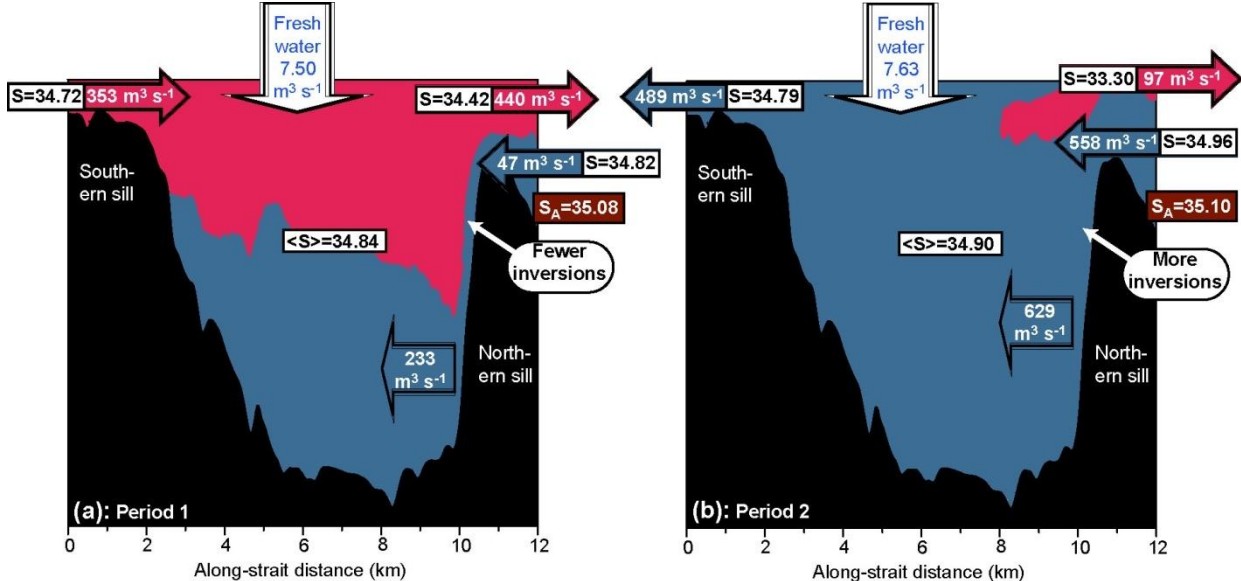

**Figure 12.** Schematic flow patterns and exchanges for the two periods defined in Fig. 7. Red and blue areas show north- and south-going (cross-estuary averaged) flow, respectively (based on Fig. 8). Horizontal arrows over the two sills show average volume transport and (transport-averaged) salinity over the sills for each period. The wide south-going arrow at depth for each period shows the total south-going volume transport just south of the northern sill (along-strait distance $\approx$ 10 km). Vertical arrows show average freshwater supply to the estuary for each period, including river and hydropower supply (constant) as well as precipitation (variable). The values for $<S>$ are the salinities averaged over the whole strait for each period. Brown boxes show the salinity, $S_A$, for the Atlantic water at depth north of the northern sill. Differences of the frequency of density inversions over the slope south of the northern sill (white arrows) are based on Fig. 11. Note that the transport values are based on averages for each period with sea level kept constant, which explains why they do not balance.

The background colours in Fig. 12 indicate net (i.e., period-wise averaged) flow through the strait. The red areas in Fig. 12a show that the upper layers have a net northward flow during Period 1. On an hourly time scale, water flows back and forth across the sills, but on average, there is a net flow from the region south of the southern sill, through the strait, and out across the northern sill. The total volume of water carried through the strait by this net flow during the 188 hours of Period 1 is slightly more than the volume of the strait. Based on this, the flushing time for the whole strait during Period 1 is 7.6 days.

During Period 2, the net flow is quite different. Now, the blue colour in Fig. 12b indicates net southward flow through almost the entire strait. Only a small near-surface region close to the northern sill has net northward flow. Even though Period 2 (158 hours) is shorter than Period 1 (188 hours), the net amount of water flowing through the strait is somewhat higher and the flushing time for the whole strait is only 5.5 days during Period 2.

The most pronounced difference between the two periods lies in the net inflow from the north and its passage through the deep parts of the strait. The net southward flow of seawater across the northern sill along the bottom was more than ten times higher during Period 2 (558 m$^3$ s$^{-1}$) than Period 1 (47 m$^3$ s$^{-1}$) and it was less diluted by the fresher waters on top. This is seen by comparing the transport-averaged salinity of the southward inflow (values in arrows) with the salinity $S_A$ of the pure Atlantic water found at depth north of the strait (values in brown boxes).

After passing over the northern sill, the dense seawater from the north tends to stay close to the bottom as it flows over the southern slope of the sill (Fig. 11d). This is the case during both periods, but more so during Period 1 (Fig. 8). During this descent, the seawater entrains and is diluted by water already in the strait. Thus, the 47 m$^3$ s$^{-1}$ that crossed the sill during Period 1 increased to 233 m$^3$ s$^{-1}$ at depth (Fig. 12a).

For Period 2, the deep arrow in Fig. 12b (showing 629 m$^3$ s$^{-1}$) represents net southward flow through the whole water column just south of the northern sill, but most of that is below sill level of the northern sill. This is evident from Fig. 9, which also illustrates that the volume transport below sill level of the northern sill and hence flushing rates of deep areas was two to three times higher during Period 2 than Period 1.

During Period 2, the seawater from the north seems to experience a higher frequency of density inversions (Fig. 11c) and to be more spread out through the water column (Fig. 8) than during Period 1. This might indicate different mixing regimes during the two periods but the model is hydrostatic and was not set up for detailed mixing studies. Thus, the simulations do not allow definite conclusions on this question.

According to the simulations, the strait, thus, switches between periods (such as Period 1) of a fjord-like behaviour with a two-layer, estuarine-type, circulation system and more strait-like periods (such as Period 2) with uni-directional

(southward) flow at all depths except for a small area over and close to the northern sill. These periods, which typically last a week each, are generated by the long-period tidal constituents of sea level variation and analyses of tide gauge data from Tórshavn and Eiði prove that they are real and not model artefacts (Fig. 10). Apparently, the variations are stronger in the model simulations than in nature, but this is difficult to assess on the basis of the available data since Tórshavn is located rather far south of the strait and amphidromic regions are characterized by large spatial changescharacterise.     These features result from a combination of topographic, freshwater input, and tidal characteristics that seem to be rather unique for this strait. During strait-like periods, such as Period 2, our system has some resemblance to other straits, such as the Menai Strait, which also experiences strong tidal forcing (Harvey, 1968; Campbell et al., 1998), but our system remains stratified (Fig. 8) rather than well-mixed and does not fit well into any of the classes in Table 1 by Li et al. (2015).

During fjord-like periods, such as Period 1, the estuarine character is more pronounced, which allows comparison to the huge literature on estuaries, both in the form of textbook or review papers (Dyer, 1997; Farmer and Freeland, 1983; Geyer and MacCready, 2014) and studies on individual estuaries/fjords. Again, there are similarities, but also differences. Thus, the circulation in fjords is often described as three-layer rather than two-layer (Dyer, 1997; Valle-Levinson et al., 2007). For our system, a two-layer circulation seems to dominate both for the overall average (Fig. 2a and Supplementary Fig. S8) and for the fjord-like Period 1 (Fig. 8), but this is also due to the chosen simulation period. If the simulation had been during a stagnation period in summer, the result would have been quite different.

The most unique character of the system studied here is, however, the periodic shifts between strait-like and fjord-like behaviour, induced by long-period tides. We have not been able to find any other region in the literature with a similar behaviour, but the signs may be rather subtle and in our case they would not have been identified without the help of the high-resolution model. Thus, there may be other areas with similar characteristics that has been overlooked.

## 5 Conclusions and recommendations

Although open in both ends – and thus by definition a strait – the water body treated in this study in many ways behaves like a fjord. Except for its southernmost parts, the strait is usually stratified with a surface layer that has reduced salinities from the large amount of freshwater entering as runoff. The stratified region includes the waters over the northern sill where brackish water flows out of the strait, mainly in the surface layer, while saline seawater flows in, mainly at depth. The southern sill is much narrower and shallower than the northern sill. This causes high velocities and vertical mixing to be generated over the southern sill by the strong tidal forcing. The high velocities and shallow conditions also cause strong bottom friction. According to the simulations, the work done by the bottom friction typically removes more than half the potential energy generated by the forcing and limits the amount of water passing southwards out of the strait. Thus, most of the water entering the strait across the northern sill during the rising tide also leaves the strait across the northern sill during the ebbing tide rather than passing through the strait.

The tidal forcing is dominated by semidiurnal sea level variations that are much stronger north of the strait than south
of it due to the proximity of an amphidromic region. These variations control the flows within and through the strait on time
scales up to a day, but the tidal forcing also includes variations with fortnightly and monthly periods, which generate slow
variations in the flow field. According to the simulations, these long-term flow variations also induce variations in the
hydrographic structure so that the strait switches between periods of a more strait-like and a more fjord-like character with
each period typically lasting one week. Associated with these switches there are pronounced changes in the flow between the
strait and the area south of it. Also the flushing rate of the deep water in the strait is affected by these switches and they need
to be taken into account for the management of aquacultural and other activities. Analyses of tide gauge measurements off
both ends of the strait demonstrate that these long-term variations in tidal forcing are real and not model artefacts. There are,
however, indications that they may be enhanced in the model simulations and it is recommended that a new simulation is
implemented together with a more targeted field experiment to provide a better foundation for evaluating the rather unique
behaviour of this strait.

*Code availability.* The source code of ROMS is available from ROMS, http://myroms.org.

*Data availability.* Landsverk data of sea level are available upon request (www.landsverk.fo). Data from DMI surface air
pressure and precipitation are available upon request (www.dmi.dk). Current measurements and CTD observations are
available upon request from SVE. The data from local energy supplier (SEV) are available upon request (www.sev.fo). The
model output from the FC32m model are available from SVE upon request.

*Author contributions.* SVE, JA and KS conceptualized the model study, JA set up the model and ran the model. SVE
performed data analysis on model data, EO performed analysis on observations. KS, EK and BH provided supervision and
guidance. SVE and BH wrote the manuscript with input from all the authors.

*Competing interests.* The authors declare that they have no conflict of interest.

*Acknowledgements.* Thanks to Lars Asplin, Institute of Marine Research, Norway, for providing the network and for the
general introduction to ROMS. This paper is part of the PhD project by SVE with EK as main supervisor. The PhD project is
financially supported by Statoil Faroes, Mowi Faroe Islands, P/F Bakkafrost and the Faroese Research Council (Grant no.
0445). We thank Manuel Diez-Minguito, Hans Burchard and two anonymous referees for very helpful comments.

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
