# Peer review of "A tidally driven fjord-like strait close to an amphidromic region"

_Ocean Science, 2021_

## Author Comment (AC1)

**Reply to comments from Referee 2 (R2) on: A tidally driven estuary close to an amphidromy by Sissal Vágsheyg Erenbjerg et al.**

We thank the referee for positive and constructive comments.

**General responses to comments from all or two of the referees**

In the reviews from the three referees, there are a number of points addressed by all or two of them. They necessitated comprehensive revisions of the manuscript and, here, we give a general overview of these points and our responses to them. For more specific comments from R2, see below.

1. One of these points is our use of the term "estuary", criticized by all the referees. We have followed the recommendation of R1 and R2 to use "strait" throughout the manuscript, instead. We still feel that this strait in many ways behaves like an estuary, but we acknowledge that this was badly motivated, especially in the Introduction. In the revised version, this question is now addressed more thoroughly in the Introduction. Other points of criticism were a too superficial treatment of the tides and also our lack of clearly stated objectives.

2. To address these points, we have re-written the Introduction completely. There, we now emphasize that the freshwater supply is sufficient to lower the salinity appreciably and that the cross-sectional area of the southern sill is so small that it only allows slightly less than half of the water entering the strait across the northern sill during flood to pass through the strait, on average. This makes the strait behave much like an estuary and motivates the new title in the revised version: "A tidally driven fjord-like strait close to an amphidromic region".

3. In the new Introduction, we also address the tidal regime more comprehensively, referring to a supplementary figure with maps of the amplitudes of the main semidiurnal and diurnal tidal constituents, based on the parent (800 m) model. We stress that the amphidromic character of the region south of the strait includes the four dominant semidiurnal and to some extent also the two dominant diurnal constituents.

4. In the literature, we have not found any water body that shares this combination of fjord-like topography (sills) and competition between freshwater and tidal forcing. In the new Introduction, we argue that this justifies a closer study even though this strait is small compared to most better-known straits. Based on this motivation, we have re-phrased the objectives and methodology of the study, hopefully to be clearer.

5. Another common point of criticism from all of the referees was in regard to model validation. We have now added a new section comparing the characteristics of the main tidal constituents as measured at two locations on either side of the strait with those in the parent (800 m) model (the southern location is not within the domain of the high-resolution model). The comparison (including the new Table 1) verifies that the parent model reproduces the dominant tidal

characteristics fairly well. We have also added a new supplementary figure with Hovmøller diagrams comparing simulated velocities in the strait with those measured by ADCPs to compare velocity profiles at intra-tidal time scales as requested by all of the referees.

6.  We also acknowledge that the lack of hydrographic observations during the modelling period and constancy of freshwater supply in the model make our attempt at validation of salinity fields in the model rather unrealistic. We have therefore moved the old Fig. 4 to the supplement and modified the text on this matter. Following the recommendation from R1, we have furthermore moved model validation from being a separate section (old Sect. 3) to a subsection in Sect. 2.

7.  As motivated in the new Introduction, we feel that the special features of this strait distinguish it from the typical strait and make it worth a study. In our opinion, the main result of the study is, however, the long-period (fortnightly and monthly) variation of the daily-averaged (25 hour) net flow through the strait, which changes systematically between northward and southward flow with periods on these time scales. When combined with the abovementioned special features, this example of long-period tidal forcing is to our knowledge sufficiently unique to justify publication in OS. Unfortunately, we have to acknowledge that we did not discuss or emphasize this message adequately. In the revised version, we have exchanged old Fig. 10 with a new figure (new Fig. 9) that better documents that this feature is not an artefact of the model, but is also to be found in the measured sea level data. We have tried to clarify this point in the new Results and Discussion sections, we have re-written the abstract to more clearly emphasize the results of the study (as recommended by R1), and we have converted the Recommendations section to a "Conclusions and Recommendations" section (as recommended by R1).

**Specific responses to comments from R2**

**Comment:** not clear how geographical names are to be spelled - please provide phonetic translation; similar comments apply to maps (Fig. 1) where mesh indices are shown instead of coordinates in meters: please present information such that it is easily accessible to be memorised and interpreted by the average reader
**Reply:** We have added phonetic translation for the relevant local names. We have also replaced the mesh indices (grid numbers) in Fig. 1 and elsewhere by geographical distance.

**Comment:** in the Introduction the study area is presented as an estuary or fjord, i.e. a land-ocean transition space, but obviously it is an ocean strait.
**Reply:** "Estuary" has been replaced by "strait" (see General Responses Bullet point 1, above).

**Comment:** Here a decent review on circulation in ocean straits is imperative, for example Danish straits and Bosporus (amongst others) have been studied well: Identify the knowledge that can be transferred from other straits to the local strait, identify the knowledge gaps and say how the gaps shall be closed using the methodology of this study.
**Reply:** In the revised version, we now refer to the review paper on shallow straits by Li et al. (2015) and list several features that distinguish our area from a typical strait (see General Responses Bullet points 2, 3, 4).

**Comment:** Model area, model validation: Why is model area so small? This creates several problems: As water level differences are substantial for the conclusions of the study, the model area should include both gauges shown in Fig. 1. Alternatively authors could validate the parent model against these gauges.
**Reply:** Doing a model study will always be a delicate balance between available computing resources, time and resolution. The present study is part of a PhD-project with main workplace in the Faroes, degree-giving university in Copenhagen, Denmark, while computations were run in Bergen, Norway, as in-kind contribution. This has put severe restrictions on computing resources. The model domain described here also has a very high resolution of 32m x 32m and even though the area covered is rather small we still have 785 x 185 horizontal grid points with 35 layers in the vertical. We have followed the advice and added a subsection on tidal validation of parent (800 m) model versus tide gauges (new Table 1).

**Comment:** Salinity validation reveals the model is too mixed - this hints at underestimated exchange flow/density driven circulation - here the area outside the sills could be crucial but it is excluded from the model which could be a serious dynamical flaw.
**Reply:** We agree that the salinity variation is underestimated by the model and too strong mixing in the model may be one reason. The large discrepancies are, however, characterized by the observed CTD profiles showing large amounts of freshwater mixed down to depths 10-30 m. This may be caused by storms that induce a lot of runoff combined with strong mixing. Since the model assumed constant

runoff, we cannot expect it to catch these events, but this ought to have been better explained in the text. We have now acknowledged that data does not really allow us to validate the salinity variations in the model, moved Fig. 4 to the supplement, and modified the text (see General Responses Bullet point 6, above).

**Comment:** Although tides are important the validation considers daily scale which does not make sense. Sub-tidal flows are usually a function of both overtides and density driven flows - it would make sense to start validation at the intra-tidal scale.
**Reply:** We have now added validation of the tidal constituents in the model (new Table 1). We have also added two figures with juxtaposed Hovmöller diagrams from ADCP and model to the Supplement and have clarified the text in the revised manuscript (see General Responses Bullet point 5, above).

**Comment:** Density inversion in a hydrostatic model - how is this possible?
**Reply:** Apparently, high-density/salinity water crosses the northern sill and flows southward where it may pass over less dense water. Rapid fluctuations of denser water above slightly lighter water then occur, and since we apply a model using the hydrostatic assumption, we will not be able to reproduce this in detail. However, the inflow of denser water from the north is probable, but in our simulations, the internal vertical mixing and turbulence parameterization will homogenize such waters leaving the southern part of the fjord system in a hydrostatic balance.

**Comment:** ll15 "results from a model" from a model simulation
**Reply:** The text has been modified

**Comment:** ll18 a sill is an elevation, maybe say depth is 4 m at the sill
**Reply:** Done

**Comment:** ll27 how large?
**Reply:** The Introduction has been changed so that this is no longer applicable (see General Responses Bullet point 2, above).

**Comment:** ll37-38 why study winter conditions when for the Stakeholders (aqua farms etc) summer dynamics are more relevant?
**Reply:** We would have preferred to have a full year of simulation as this would help us describe the annual cycle of water exchange in the fjord. Unfortunately, the access to computational resources was quite limited. We therefore started with the winter or more "normal" conditions to get a picture of the best case scenarios of water exchange in the fjord. In the new Introduction, this is hopefully better motivated. The aquaculture farms will also benefit from better information about flow during the winter, especially as regards transmission of parasites (salmon lice) between farms. Here, the suggested long-period (fortnightly and monthly) variations of net flow may perhaps be developed into a useful management tool.

**Comment:** ll62-66 Please provide a consistent description of the aims of the study, list the research questions

**Reply:** This is hopefully better in the revised version (see General Responses Bullet point 4, above).

**Comment:** Section 3: Why not use the ADCP data to illustrate intra-tidal dynamics, validate simulated currents?

**Reply:** Is now done in the new Table 1 and a new supplementary figure (see General Responses Bullet point 5, above).

**Comment:** In estuaries per definition river flow affects the salinity field, and Fig. 4 shows that freshwater input is probably significant

**Reply:** It is not quite clear to us, what the referee intended with this comment, but Fig. 4 has in any case been moved to the supplement (see General Responses Bullet point 6, above).

**Comment:** Fig. 2a: what is the data basis for the red and blue colors - model or observations? Specify in caption.

**Reply:** Done

**Comment:** Fig. 3: Consider illustration and validation on intra-tidal scales...consider showing these numbers in a table instead

**Reply:** Validation of tidal constituents and intra-tidal variations has been added (see General Responses Bullet point 5, above). Originally, we also tried to put the information in Fig. 3 into a table, rather than a figure, but found it difficult to show the same overall information.

**Comment:** ll247 no hydrographic observations during the simulation period" - what about using climatological data

**Reply:** This is what we tried to do in Fig. 4, although we admit that our climatological data set is not perfect. For the parent (800 m) model, a fairly comprehensive comparison was made between observed and modeled hydrography in a previously published manuscript. This ought to have been referred to in this text and we have now done that.

**Comment:** ll291 please specify what is meant by highly non-linear flows.

**Reply:** This was not well phrased. The sentence is hopefully clearer in the revised manuscript.

**Comment:** tidally-rectified currents: probably a tidal analysis of ADCP data and model results can be very helpful in this case

**Reply:** This explanation referred to processes outside the model domain, but we agree that it was probably too speculative and we have deleted this paragraph.

---

## Author Comment (AC2)

**Reply to comments from Manuel Diez-Minguito (R1) on: A tidally driven estuary close to an amphidromy by Sissal Vágsheyg Erenbjerg et al.**

We thank the referee for positive and constructive comments.

**General responses to comments from all or two of the referees**

In the reviews from the three referees, there are a number of points addressed by all or two of them. They necessitated comprehensive revisions of the manuscript and, here, we give a general overview of these points and our responses to them. For more specific comments from R1, see below.

1.  One of these points is our use of the term "estuary", criticized by all the referees. We have followed the recommendation of R1 and R2 to use "strait" throughout the manuscript, instead. We still feel that this strait in many ways behaves like an estuary, but we acknowledge that this was badly motivated, especially in the Introduction. In the revised version, this question is now addressed more thoroughly in the Introduction. Other points of criticism were a too superficial treatment of the tides and also our lack of clearly stated objectives.

2.  To address these points, we have re-written the Introduction completely. There, we now emphasize that the freshwater supply is sufficient to lower the salinity appreciably and that the cross-sectional area of the southern sill is so small that it only allows slightly less than half of the water entering the strait across the northern sill during flood to pass through the strait, on average. This makes the strait behave much like an estuary and motivates the new title in the revised version: "A tidally driven fjord-like strait close to an amphidromic region".

3.  In the new Introduction, we also address the tidal regime more comprehensively, referring to a supplementary figure with maps of the amplitudes of the main semidiurnal and diurnal tidal constituents, based on the parent (800 m) model. We stress that the amphidromic character of the region south of the strait includes the four dominant semidiurnal and to some extent also the two dominant diurnal constituents.

4.  In the literature, we have not found any water body that shares this combination of fjord-like topography (sills) and competition between freshwater and tidal forcing. In the new Introduction, we argue that this justifies a closer study even though this strait is small compared to most better-known straits. Based on this motivation, we have re-phrased the objectives and methodology of the study, hopefully to be clearer.

5.  Another common point of criticism from all of the referees was in regard to model validation. We have now added a new section comparing the characteristics of the main tidal constituents as measured at two locations on either side of the strait with those in the parent (800 m) model (the southern location is not within the domain of the high-resolution model). The comparison (including the new Table 1) verifies that the parent model reproduces the dominant tidal

characteristics fairly well. We have also added a new supplementary figure with Hovmøller diagrams comparing simulated velocities in the strait with those measured by ADCPs to compare velocity profiles at intra-tidal time scales as requested by all of the referees.

6. We also acknowledge that the lack of hydrographic observations during the modelling period and constancy of freshwater supply in the model make our attempt at validation of salinity fields in the model rather unrealistic. We have therefore moved the old Fig. 4 to the supplement and modified the text on this matter. Following the recommendation from R1, we have furthermore moved model validation from being a separate section (old Sect. 3) to a subsection in Sect. 2.

7. As motivated in the new Introduction, we feel that the special features of this strait distinguish it from the typical strait and make it worth a study. In our opinion, the main result of the study is, however, the long-period (fortnightly and monthly) variation of the daily-averaged (25 hour) net flow through the strait, which changes systematically between northward and southward flow with periods on these time scales. When combined with the abovementioned special features, this example of long-period tidal forcing is to our knowledge sufficiently unique to justify publication in OS. Unfortunately, we have to acknowledge that we did not discuss or emphasize this message adequately. In the revised version, we have exchanged old Fig. 10 with a new figure (new Fig. 9) that better documents that this feature is not an artefact of the model, but is also to be found in the measured sea level data. We have tried to clarify this point in the new Results and Discussion sections, we have re-written the abstract to more clearly emphasize the results of the study (as recommended by R1), and we have converted the Recommendations section to a "Conclusions and Recommendations" section (as recommended by R1).

**Specific responses to comments from Manuel Diez-Minguito**

**Comment:** I rather focus the abstract on physics, processes, etc. more than the model implementation itself. (The same occurs elsewhere in the manuscript, mainly in introduction, discussion and conclusions.)
**Reply:** The abstract has been re-written to better present the results of the study rather than the model (see General Responses Bullet point 7, above). We have also tried to do the same re-focusing throughout the manuscript.

**Comment:** L1. 'describes the implementation'. I think this manuscript does more than that.
**Reply:** We have re-written the abstract. See comment above.

**Comment:** L2. Not sure to call this 'estuary', as it is not semi-enclosed body of water. It wouldn't be more like a 'strait'?
**Reply:** We now use the term "strait" rather than "estuary" to refer to the study area (see General Responses Bullet point 1, above).

**Comment:** L9. 'Surprising'. I suggest to ommit this kind of valorative adjectives. (here and elsewhere)
**Reply:** Has been done.

**Comment:** L1 and L12. Rephrase "describes the implementation...". "We recommend that..."
**Reply:** These phrases are not in the new abstract, which has been re-written.

**Comment:** L29. 'estaury'. Correct here and elsewhere (L30, L38,...)
**Reply:** The term "estuary" has been replaced by "strait" in most places. Where it still occurs, it should be correctly spelled.

**Comment:** L30. Please indicate in which ways.
**Reply:** In the revised version, the term "typical estuary" has been replaced by "typical shallow strait" in the first paragraph of the Introduction, which is followed by a number of distinguishing features.

**Comment:** L37-38. Please describe what are "normal conditions" are. Are those described in the next paragraph? What is "non-sill estuary circulation"?
**Reply:** The terms "normal conditions" and "non-sill estuary circulation" are no longer in the manuscript.

**Comment:** L46. Perhaps would be convenient to show the location of the amphidromic point in a Figure.
**Reply:** In the new Introduction, we now discuss the tidal regime more comprehensively and include a supplementary figure (see General Responses Bullet point 3, above).

**Comment:** L50. "To understand how these various forcing mechanisms affect mixing and circulation within and out of the estuary" This is the objective, am I right? Please state it clearly. The use of one numerical model or another is part of the methodology.
L60-61. Is this also part of the objectives? General objective, perhaps?
L50-66. I suggest the authors to reorganize this part of the introduction. General and specific objectives/aims should be clearly and logically stated. And then, describe how are they addressed (methods, numerical model).
**Reply:** The Introduction has been completely re-written and should now better state objectives and methods clearly.

**Comment:** L83. I do not understand this. Does this mean that the daily input to ROMS is 1.7e+8/365 m3/day and 6.3e+7/365 m3/day?
**Reply:** has been clarified in the new text.

**Comment:** L85. Is it enough one day to spin-up the model? How do you determine that?
**Reply:** The high-resolution (32 m) model starts two weeks after the 160 m model, which starts four weeks after its parent (800 m) model. Since the 160 m model has many mesh points within the strait, the starting conditions for the 32 m model should be approximately realistic. This is verified by inspection of the temporal evolution of parameters (especially kinetic energy) during the start period of the 32 m model.

**Comment:** I suggest the authors to move Section 3 to Section 2.1. After all, model calibration and validation is part of the set-up process.
**Reply:** The old Section 3 has been moved to be a subsection in the new Section 2 (although not 2.1).

**Comment:** L98. Averages for the whole sampling period? Monthly? Please indicate the time span in which averaged are performed. Also in Figure 2 and 3.
**Reply:** Has been done.

**Comment:** L101-102. Please explain briefly the 'modeling details'.
**Reply:** Has been re-written.

**Comment:** Please consider to show one or two panels that show time series of observations confronted to model output.
**Reply:** We have added a supplementary figure for this purpose (see General Responses Bullet point 5, above).

**Comment:** Last two paragraphs in section 3.2 may shed doubts if the model is correctly validated in terms of salinity. Should not salinity variations in the shelf and freshwater discharges time series be included as input in the model?
**Reply:** We agree that our treatment of salinity validation was too superficial and that we do not have the data for adequate validation of simulated salinity. This question should now be better treated in the revised manuscript.

**Comment:** L132-134. Move these lines to methods part.
**Reply:** Has been done.

**Comment:** L135-. Too late, I think. Please mention this or move the paragraph above.
**Reply:** This paragraph has been modified and moved to Sect. 2.

**Comment:** L140. I think Results section should start here.
**Reply:** In the revised version, it does so.

**Comment:** L142. I guess qN and qN are cross-sectionally integrated. Please indicate. Are tidal variations in the sea level (elevations) also considered or only the mean cross-section?
**Reply:** Yes, they are cross-sectionally averaged and sea level variation has been included. This now more clearly stated in the manuscript.

**Comment:** L146. How the one-hour lag compares with the tidal wave celerity?
**Reply:** We have now added a comment on this to the discussion of lags in the Discussion section (beginning of old Sect. 5.2).

**Comment:** L147. Not always. There are some intervals in which the volume flux is unidirectional both during ebb and flood.
**Reply:** This sentence has been changed to "Most of the time, the volume transport changes sign four times a day, as would be expected with semidiurnal tidal forcing, although there are a few cases with unidirectional flow lasting more than a day".

**Comment:** L159-161. Table 1. The cross-correlation between qS and qN has a 1 hour lag. If the barotropic pressure gradient is driving the volume flux, why there is no lag in the crosscorrelation between qS and (hN-hS)? Any clue? Perhaps there is a baroclinic mode which is also driving the net volume flux in the estuary?
**Reply:** With 1-hour interval in the model data, the difference between 1-hour lag and 0-hour lag may in reality be only a difference of a few minutes (e.g., between 31 minutes and 29 minutes). We have added some text on this to the discussion of lags in the Discussion section (beginning of old Sect. 5.2).

**Comment:** L165. 'Speed' vS. Cross-sectionaly averaged velocity?
**Reply:** Since the volume transport varies linearly with speed while the kinetic energy varies with the squared speed, the model is based on the assumption that the speed does not vary spatially on the cross-section at each instant. We have changed the text "assumed to be homogeneous on the section" to "assumed not to vary spatially on the cross-section". We have also added some text to justify this assumption.

**Comment:** L167. Assuming that there is no correlation between A and vS.
**Reply:** We have added that A is assumed constant.

**Comment:** L169. It would be nice to test the sensitivity of the fit to the locations of hS and hN.
**Reply:** We have now redone this analysis, varying the locations defining ΔhS to minimize the parameter δ. This required updating the old Fig. 6 and the old Table 1. We have also added an equation to make the point clearer.

**Comment:** L170. I don't understand. Friction is always there. You mean when friction is included in Bernoulli equation?
**Reply:** We have modified the text and added an equation to clarify this point

**Comment:** L175. This is quite usual in many estuaries.
**Reply:** We agree that fortnightly variations in the amplitude of inflow to and outflow from an estuary are common (e.g., as a superposition of $M_2$ and $S_2$), but over 25 hours they will approximately balance. Here, we have an example of the net (25-hour averaged) volume transport through the strait varying on fortnightly (and monthly) time scales. We do not know of any similar example in the literature. We see this as the most interesting result of our study, but we acknowledge that this was not adequately addressed in the original manuscript (see General Responses Bullet point 5, above). In the revised version, we have expanded the treatment of this point in the new Discussion section. The text has been modified to clarify.

**Comment:** Figure 7. Define the daily averages <...> in the main text.
**Reply:** Has been done.

**Comment:** is consistently lower in magnitud than . Why? I suggest to plot in the same panel qN, qS, hN, hS and their differences. It seems that and are slightly out of phase?
**Reply:** Some text (parameters ?) seems to be missing in the referee comment and it is not clear to us, what the referee asks. We have made a new figure as suggested and added it to the supplement.

**Comment:** Figure 8. Indicate that the horizontal axes is along-estuary. Label northern and southern sills (or simply S and N). Instead of "grid numbers", use km.
**Reply:** Has been done.

**Comment:** L192-193. Denser water? What are the mixing rates?

**Reply:** We now note that this is denser water that manages to descend before losing its excess density due to mixing.

**Comment:** L194-196. Although it seems plausible, it would be good to provide mixing rates to support this statement.

**Reply:** We have modified the text to be more concise.

**Comment:** L201-202. Please clarify this. Shouldn't be a sign of this also in the salinity color map?

**Reply:** We acknowledge that this was too strongly phrased. The text has been modified.

**Comment:** L187-207. Overall, there are some unsupported statements here. In my opinion, authors' arguments on mixing, vertical water movements, effect of discharges, etc. do not seem to be supported by any data. Could the authors provide additional numerical evidences? (mixing rates, vertical and lateral velocities?).

**Reply:** In the revised version, this text has been modified and we have tried to be more careful not to present unsupported statements.

**Comment:** Figure 9. Is this figure really necessary?

**Reply:** We believe that this figure has an important message about the difference between the two periods, but we have probably not explained the message adequately. In the new version, we have retained the figure, but modified the text substantially.

**Comment:** L212-214. Both time series can be corrected for the barometric effect. Figure 10. I guess panel a conveys the same information as correlations in Table 1. I suggest to remove panel a. (or substitute Table 1 by lagged cross-correlation plots)

**Reply:** The old Fig. 10 and the associated analysis have been replaced by a new figure showing the importance of the long-period tides.

**Comment:** L223-241. This is interesting, but I think it deserves to be much more elaborated. At which depths occur the inversions? Above or below the picnocline? More reliable/informative (but still simple) measurements for stratification could be Brunt-Vaisala frequency or potential energy anomaly. How they would compare with the turbulent kinetic energy? How much mixing induce then these inversions? Do you think this inversions could have a (baroclinic) influence on the net exchange flow?

**Reply:** We agree that this ought to be more elaborated, but unfortunately the model was not set up with a view to study mixing and parameters such as turbulent kinetic energy or dissipation rate were not stored. In the revised version, we have therefore restricted the treatment to note that these inversions occur in a systematic manner and that they ought to be further studied.

**Comment:** L247. Please discuss the consequences of it.

**Reply:** The old Sect. 5.1 on model performance has been fundamentally re-written (see General Responses Bullet points 5 and 6, above).

**Comment:** L253. As I mentioned above, it would be nice to see where the amphidromic point is. Also, I would expect that the tidal wave propagates faster around Faroe Islands than through the fjord. Is it so? If yes, there would be probably a superposition of both waves at a certain location near the south sill. Have you looked into this?
**Reply:** As noted in the General Responses Bullet point 3, above, we have added text on the tidal regime to the Introduction and a figure to the supplement. We have not tried to analyze the tidal wave propagation over the Faroe shelf, although that would be an interesting study.

**Comment:** L257-258. I think the authors analysis here is based on the Bernoulli equation. If I'm not wrong, the determination of gamma only indicates that friction is overall important, but not that friction occurs mainly in the southern sill. Could you elaborate more on that, please?
**Reply:** We have added some text referring to the high speeds (old Fig. S6) to argue, why much of the loss probably is close to the southern sill.

**Comment:** L258-260. It is always so, I guess. Not sure what the reader should conclude from this statement...
**Reply:** This statement has been removed.

**Comment:** L265. If the amphidromic point is located near Tórshavn, why the tidal wave enters the estuary from the north (L252)? Notice that this is about timing, not amplitudes.
**Reply:** Much of the information on tides has been moved to the Introduction and the text on L265 has been modified.

**Comment:** L265-269. For me this is something that should have been mentioned in the introduction, not in the discussion section. This would have helped to better undertand the qN and qS variability.
**Reply:** Has been done in the revised version (see General Responses Bullet point 3, above).

**Comment:** L276-285. A harmonic analysis of sea levels could shed some light on this. Probably a mutual non-linear interaction between M2 and S2 produces a fortnightly compound tide MSf, which is (almost) in phase with the spring-neap tidal cycle.
**Reply:** We now include harmonic analyses of both simulated and observed sea level and the discussion on this topic has been substantially modified with more emphasis on the long-period tidal constituents (MM, MSM, MF, and MSf).

**Comment:** L290. Again, computation of mixing rates and TKE along the estuary (or better computation of terms of the momentum) could support this.
**Reply:** As noted above, the model was unfortunately not set up to store the relevant parameters for this.

**Comment:** L311-317. I think this would be easy to check out by comparing (observed or modelled) elevations and currents at both sills.

**Reply:** We believe that the referee has misunderstood our text. The text has been re-written for clarification.

**Comment:** L338. What is then the flushing time estimated from this? (The same in L341)
**Reply:** These numbers have been added.

**Comment:** L344. There is no two layer circulation in the southern sill. However, the two layer circulation in the northern sill persists during both periods, altough it varies in magnitude. I think this is remarkable. It would be very interesting to comment a bit more on this. Is there any sign of this "fortnightly pumping" in the shelf, out of the estuary? Or inside the estuary, in its deep waters? Maybe because of this "fortnightly pumping", stagnant conditions are not observed during winter? How is the circulation during summer? Is this pumping effect also present when stagnant conditions near the bottom are observed? Are there other fjords or straits that show similar circulation patterns? BTW, please consider to put the study in a wider (global) context. Most references are local.
**Reply:** The difference in stratification over the two sills is one of the reasons that we consider this strait to act like an estuary. We have added some text to this effect. We have not looked for the "fortnightly pumping" on the shelf, but for the deep water in the strait, there is a marked effect, which is what we wanted to illustrate with the old Fig. 9. We have tried to emphasize this point more clearly now. Since this "fortnightly pumping" is tidally driven (better justified in the revised version), we do not expect a large seasonal cycle. No other Faroese fjord or strait has the features that make this effect so pronounced in our study area, so we would not expect to see such a clear signal anywhere else in the Faroes. And, something similar may be said more globally. We have not been able to find any other fjord-like strait with this "fortnightly pumping" in the literature, which is also why we have not put more effort into putting our study into a global context. Perhaps, there are similar systems elsewhere, where this behavior has not been identified. We probably would not have noted this for our study area without the high-resolution model.

**Comment:** L351-. Again I find mixing discussions somehow "loose"
**Reply:** We must acknowledge that this is probably correct and have tried to make the text on mixing more concise.

**Comment:** Why not Conclusions instead of recommendations? I suggest to rewrite Section 6 to frame it properly as Conclusions.
**Reply:** Has been done.

---

## Author Comment (AC3)

**Reply to comments from Referee 3 (R3) on: A tidally driven estuary close to an amphidromy by Sissal Vágsheyg Erenbjerg et al.**

We are of course sorry about the overall verdict (rejection) of R3. We still feel that the results of our study are sufficiently special and interesting to justify publication in OS, especially the fortnightly variation in net flow through this strait and its effect on the fjord-like circulation. We acknowledge, however, that this was not well emphasized in the original text and that one might have to read the manuscript very thoroughly to get the full impact. In the revised version, we have tried to rectify this and both the abstract and Introduction are more or less completely re-written. In spite of the overall verdict, R3 has several constructive and useful comments, for which we thank her/him.

**General responses to comments from all or two of the referees**

In the reviews from the three referees, there are a number of points addressed by all or two of them. They necessitated comprehensive revisions of the manuscript and, here, we give a general overview of these points and our responses to them. For more specific comments from R3, see below.

1. One of these points is our use of the term "estuary", criticized by all the referees. We have followed the recommendation of R1 and R2 to use "strait" throughout the manuscript, instead. We still feel that this strait in many ways behaves like an estuary, but we acknowledge that this was badly motivated, especially in the Introduction. In the revised version, this question is now addressed more thoroughly in the Introduction. Other points of criticism were a too superficial treatment of the tides and also our lack of clearly stated objectives.

2. To address these points, we have re-written the Introduction completely. There, we now emphasize that the freshwater supply is sufficient to lower the salinity appreciably and that the cross-sectional area of the southern sill is so small that it only allows slightly less than half of the water entering the strait across the northern sill during flood to pass through the strait, on average. This makes the strait behave much like an estuary and motivates the new title in the revised version: "A tidally driven fjord-like strait close to an amphidromic region".

3. In the new Introduction, we also address the tidal regime more comprehensively, referring to a supplementary figure with maps of the amplitudes of the main semidiurnal and diurnal tidal constituents, based on the parent (800 m) model. We stress that the amphidromic character of the region south of the strait includes the four dominant semidiurnal and to some extent also the two dominant diurnal constituents.

4. In the literature, we have not found any water body that shares this combination of fjord-like topography (sills) and competition between freshwater and tidal forcing. In the new Introduction, we argue that this justifies a closer study even though this strait is small compared to most better-known straits. Based on this motivation, we have re-phrased the objectives and methodology of the study, hopefully to be clearer.

5. Another common point of criticism from all of the referees was in regard to model validation. We have now added a new section comparing the characteristics of the main tidal constituents as measured at two locations on either side of the strait with those in the parent (800 m) model (the southern location is not within the domain of the high-resolution model). The comparison (including the new Table 1) verifies that the parent model reproduces the dominant tidal characteristics fairly well. We have also added a new supplementary figure with Hovmøller diagrams comparing simulated velocities in the strait with those measured by ADCPs to compare velocity profiles at intra-tidal time scales as requested by all of the referees.

6. We also acknowledge that the lack of hydrographic observations during the modelling period and constancy of freshwater supply in the model make our attempt at validation of salinity fields in the model rather unrealistic. We have therefore moved the old Fig. 4 to the supplement and modified the text on this matter. Following the recommendation from R1, we have furthermore moved model validation from being a separate section (old Sect. 3) to a subsection in Sect. 2.

7. As motivated in the new Introduction, we feel that the special features of this strait distinguish it from the typical strait and make it worth a study. In our opinion, the main result of the study is, however, the long-period (fortnightly and monthly) variation of the daily-averaged (25 hour) net flow through the strait, which changes systematically between northward and southward flow with periods on these time scales. When combined with the abovementioned special features, this example of long-period tidal forcing is to our knowledge sufficiently unique to justify publication in OS. Unfortunately, we have to acknowledge that we did not discuss or emphasize this message adequately. In the revised version, we have exchanged old Fig. 10 with a new figure (new Fig. 9) that better documents that this feature is not an artefact of the model, but is also to be found in the measured sea level data. We have tried to clarify this point in the new Results and Discussion sections, we have re-written the abstract to more clearly emphasize the results of the study (as recommended by R1), and we have converted the Recommendations section to a "Conclusions and Recommendations" section (as recommended by R1).

**Specific responses to comments from R3**

**Comment: I** am irritated about denoting the sound between the two islands as an estuary although you even write in the introduction that this is not an estuary in the classical sense. It would be better to call this here as a "narrow sound with strong freshwater run-off" or so.
**Reply:** We now use the term "strait" throughout as suggested by R1 and R2 and have changed the title (see General Responses Bullet points 1 and 2, above).

**Comment:** Here the major runoff (from a hydro power station) occurs at the open end of the sound and thus it can be expected that the general behaviour is much different than in an estuary where the freshwater run-off occurs at the closed end.
**Reply:** As we have now stressed in the new Introduction, the naturally occurring freshwater runoff (excluding the contribution from the hydropower plant) is sufficient to lower the salinity appreciably and give a salinity distribution that looks quite fjord-like (e.g., the old supplementary Fig. S3).

**Comment:** In fjords with sills, one of the major topic is the ventilation and renewal of the deep water. Since we have here a sound that is bounded by two narrow straits with sills, there is a large body of deep water residing near the bottom. To my opinion, it is a major limitation of this study that this topic is not discussed.
**Reply:** We agree that a longer simulation period would have been preferable, but the computing resources available to the study were limited and a study of deep-water stagnation would have had to include a substantially longer simulation period.

**Comment:** What are the deep water renewal processes? Is it tides or wind or surges? How often does it happen? The model system used here should be able to reproduce those dense water overflows.
**Reply:** We have tried to address this topic and noted that this process also is affected by the fortnightly variations, see old Figs. 8 and 9, but this should have been better emphasized in the text, which it hopefully is in the revised version.

**Comment:** Just initialiing the salinity and temperature fields at some instant of time and simulating for a short period might completely miss the dynamics. Here, just one day is used to let the model adjust to the initial fields, a time span that should be by far shorter than the deep water renewal time.
**Reply:** The 32m model is nested within a 160m model, which was run for two weeks before start of the 32m model and the parent (800m) model was run for four weeks before the start of the 160m model. The 160m model has many points within the estuary and we expect both upper and deeper layers to be approximately spun-up by the start of the 32m model. Also, no spin-up effects are seen in the 32m model. This has now been better explained in the revised version.

**Comment:** As for the validation, the results are very poor. Tidally resolved velocity measurements are not compared to model results, and the comparison between simulated and observed residual velocity profiles is very bad. Salinity observations are not available during the simulation period. A comparison to observed salinity profiles obtained during several other years is made, shows big differences to the model results and a high variability (and makes no sense anyway).

**Reply:** We now have included validation of tidal constituents and added a supplementary figure showing tidally resolved velocities, but the lack of observed near-surface velocities (which are stronger than the deep currents) makes this less interesting. We have also modified the text on salinity validation to admit that we do not have the data to do such a validation satisfactorily (see General Responses Bullet points 5 and 6, above).

**Comment:** With this, the model results are nit validated at all, and do probably not reflect the dynamics of the sound under consideration.

**Reply:** This is in our opinion an exaggeration. Although not perfect, some of the relationships in the old Fig. 3 are sufficiently significant to support the model results. With the added content in the revised version, this is strengthened.

**Comment:** The paper is lacking motivation. In the introduction, a clear scientific problem needs to be presented on the background of the state of the art. Here, however, very little state of the art is given, a problem is not clearly identified and hypotheses are not offered.

**Reply:** We agree that the Introduction (and abstract) were not sufficiently informative and concise. This is hopefully better in the revised version.

**Comment:** 3: Specify for which partial tide you have the amphidromic region. You probably mean the M2 tide, but please specify.

**Reply:** In the Introduction, we now emphasize that the amphidromic character applies for all (four) the dominant semidiurnal constituents and also to the two dominant diurnal constituents at least partly. We have also added a supplementary figure to illustrate this (see General Responses Bullet point 3, above)..

**Comment:** 5: I would prefer "volume transport", because I think "flux" is reserved for "transport per unit area".

**Reply:** "Volume flux" has been replaced by "Volume transport" throughout the manuscript.

**Comment:** 9: How can you verify transports with sea level observations?

**Reply:** The original text did say that (modeled) "variations in sea level differences" (not transports) were verified by observed sea level variations.

**Comment:** 10/11: reformulate this as a sentence.

**Reply:** This text has been modified in the new abstract.

**Comment:** 26: "reducing the runoff into the southern part of the sound while the northern part has received more freshwater". What is the mechanism here and how do you know?
**Reply:** The water supply to the hydropower plant is partly through tunnels that redirect water that would have gone into the southern part so that it enters the northern part instead.

**Comment:** 29/30: typo "estaury", here and at many other locations.
**Reply:** Corrected.

**Comment:** 37/38: "when the circulation is more similar to that of a non-sill estuary": At this point the reader has no idea of the salinity distribution in this sound. A better motivation is needed for the choice of the winter for this case study.
**Reply:** This statement has been removed from the revised version and the motivation clarified.

**Comment:** 39-45: The review article by Farmer and Freeland does actually discuss tides as an important process of fjord dynamics (see their section 4).
**Reply:** We no longer refer to Farmer and Freeland in the revised version.

**Comment:** 48-49: You can also have "a strong periodically varying barotropic pressure gradient through the estuary" when the amphidromic points are far away. So, at this point I do not see any special influence of the proximity of the amphidromic point apart from the fact that the M2 tide is weak.
**Reply:** This is correct, but in our case, it is the proximity of the amphidromic region that creates the large sea level differences between both ends of the strait and the strong tidal currents.

**Comment:** 56-61: I would move this paragraph to the "Materials" section, since the introduction should serve more general purposes and introduce the problem, give hypotheses, etc.
**Reply:** Since the model is the main method used in this study, we feel that it should be mentioned in the Introduction, although we have moved some of the details to Sect. 2.

**Comment:** 64/65: "One aim of this study was therefore to validate the model against these observations.": This is not a sufficient aim for a study to be published in a peer-reviewed international journal. Also the next sentence is not sufficient as motivation.
**Reply:** We have now emphasized that model validation is a secondary aim of the study. The main aim and its motivation should also be clearer now.

**Comment:** 82/83: Could you also give the runoff in m3/s which is more common.
**Reply:** has been done.

**Comment:** 83: What do you mean with constant daily run-off? I suppose that the run-off has to be given a every barotropic model time step which is much shorter than one day.
**Reply:** The text has been clarified.

**Comment:** 84/85: Not clear how the spin-up of the model can be as short as one day. How are the initial conditions for the high-resolution simulation been initiated? I guess from the level-2 nest. This needs to be explained. Since the residence time of the deep water in the sound must be much longer than one day, I wonder how good the quality of the initial condition is. Have they been validated by observations?

**Reply:** As explained in the response to a previous comment, the parent models have been spun up over much longer time and no spin-up effects were seen in the run of the 32m model. Unfortunately, we do not have observations for validating the initial conditions.

**Comment:** 96-113: I do not see any agreement between observed and simulated velocity profiles. The model results show a residual flow that is directed northwards, but the observations do not show that at all. I find it also strange to report on a study of tidal flow, have tidal flow observations at hand, but state that "a model-observation comparison of instantaneous velocities is not very meaningful". The key issue in tidal simulations is to reproduce tidal phases and amplitudes. This requirement is not met here.

**Reply:** Our intention with the quoted sentence was to emphasize that a point-to-point correspondence between model and observations requires that the phases of the tidal constituents are accurately simulated, whereas this may not be necessary to simulate the processes in the model adequately (except for the exact timing). But, we agree that this was not well phrased and this text has been modified. We also now include validation of tidal constituents (new Table 1) and have added a figure with Hovmøller diagrams (see General Responses Bullet point 5, above).

**Comment:** 115-130: Simulated salinity is here compared to observations that have been made outside the simulation period. Since salinity at the bottom should vary substantially with deep water renewal events, any similarity between observed and simulated salinity would be pure random. With this, no validation of the salinity field has been made. I wonder, if the bottom-mounted ADCP's should have included a CTD such that at least bottom salinity and temperature could be validated.

**Reply:** The salinity observations used for validation were from winter (see caption for old Fig. 4) and during that season the deep water renewal is continuous (not in events) on daily time scales or longer although variable (e.g., old Fig. 12), but we agree that our salinity validation lacks observational data to be satisfactory.

**Comment:** I am stopping here with my detailed review, since I do not think that it makes sense to deeply analyse results of a non-validated model.

**Reply:** As previously mentioned, we find this to be an exaggeration. We also note that there are many model studies to be found in the literature with little validation because adequate observations are not available, as is the case for our study.

---

## Author Response (AR2)

We thank Hans Burchard and an anonymous referee for constructive comments.

In the revised version, we have added one figure (the new Fig. S7) to the Supplement. In addition, we have revised the text of the main manuscript based on the referee comments as elaborated below and as documented by the Track Changes in the revised manuscript.

"Original line numbers" refer to line numbers in the manuscript submitted after the first revision. "New line numbers" refer to line numbers in the revised Track Changes manuscript in this submission.

**Referee 1 (Hans Burchard)**

**Comment:** 205-206: Not sure what this two-line subsection serves for. Shouldn't it be integrated into the relevant subsection related to it?
**Response:** These lines (Sect. 2.4) have been deleted and the reference to statistical significance of correlation moved to the caption of Fig. 3 (new lines 193-194).

**Comment:** 351: "result" instead of "come"?
**Response:** The suggested change has been made (new line 363).

**Comment:** 536: What do you mean with "normally stratified"? Maybe "stably stratified"?
**Response:** The word "normally" has been changed to "usually" (new line 568).

**Referee 2**

**Comment:** Still it is not clear to me why the model has not been run with more realistic or simply higher freshwater discharge? Authors write that discharge into the strait is large but somehow did not prescribe this amount.
**Response:** It is not quite clear to us, what the referee refers to here. The freshwater discharge was prescribed to the model as stated on the original lines 119-121 (new lines 119-121). Perhaps the problem is our use of the annually averaged discharge, rather than a value more appropriate for the season simulated or a variable discharge rate. We fully agree that it would have been advantageous to run the model with different discharge rates or even variable discharge. Unfortunately, limited access to computing resources prevented that. One benefit of our choice of the annual average is that it ought to represent typical conditions in the strait throughout the year, except for periods during summer with a stagnant bottom layer.

**Comment:** which is why validation ignores the surface layer.

**Response:** Again, we are not certain, how to interpret this comment, but we assume that it refers to the velocity validation (e.g., Fig. 2b,c and Supplementary Fig. S4). The reason for excluding the surface layer in this validation is that the measurements were made with bottom-mounted ADCPs, which cannot measure the surface layer due to side-lobe reflection from the surface. Certainly, the ADCPs should have been able to profile closer to the surface than they did, but the depth range used for validation was the maximum allowed by the quality controlled ADCP data sets made available to us.

**Comment:** To be acceptable for publication I expect this paper to include extensive comparison and co-interpretation with other works and concepts. Sure the study area is unique in certain ways but due to the large uncertainty in the model simulations one cannot be sure if this is an artefact. For example, the typical Fjord circulation has been described as three-layered (e.g. Valle-Levinson et al. 2007) while the estuarine circulation is typically two-layered (Geyer & MacCready, 2014).

**Response:** We have added some text on this to the discussion (new lines 552-563) where we try to provide more detailed comparison with literature on straits and fjords and briefly discuss the distinction between two- and three-layered fjord circulation.

**Comment:** Since the line of argumentation concentrates on tides, I wonder why tidal reflection at the southern sill of the Sundini strait is not discussed? It is quite improbable that all the kinetic energy is lost to mixing there, see for example Sohrt et al., 2021 discribing tidal reflection in a similar (though shallower) setting.

**Response:** The referee is probably correct that we should have mentioned tidal reflection. This is now done (new lines 247-249) where we argue that our strait (in contrast to the examples discussed by Sohrt et al., 2021) is so short in relation to the wavelength of the tidal wave that this cannot explain our result. As to the question of kinetic energy loss, we did not intend to imply that this loss was due to mixing within the water column. This misunderstanding is probably due to our unfortunate phrasing in the conclusions section (original lines 540-542). We have now modified that text (new lines 572-573). Our argument was rather that strong bottom friction over the southern sill limits the amount of water that crosses the sill. We have modified the text in the results section to make this clearer (new lines 249-250). We have also added a conceptual model (new Supplementary Fig. S7), which allows an independent (of ROMS) estimate of the energy loss due to friction (new lines 286-292). The conceptual model supports the results from ROMS, but also highlights the importance of choosing an appropriate value for the drag coefficient in the ROMS model. This is now better emphasized in the discussion section (new lines 446-452).

**Comment:** In the Discussion now authors present further results like an estimation of the flushing time. My recommendation is to link their study to other works and more specifically the general concepts presented in these works. Reference list is now quite short and mainly contains references to methods and previous works of the co-authors (not all of them in english language!).

**Response:** At the end of the discussion, we have added some text with more detailed comparison with other studies (new lines 552-563).

**Comment:** In the reply to the referees' comments I missed direct answers to the questions. It is not enough to say "we did this, see the text". You need to either say what you improved directly or indicate where the change is to be found in the revised manuscript.

**Response:** Regretfully, we must agree that our previous response was not adequate in this respect. We hope that this is better now.

**Comment:** The language needs some improvement: There are many statements in quotation marks, many repetitions of certain word like 'veryfied' and often statements like "one cannot expect" which reads a little awkward. Please try to use objective and plain language.

**Response:** Most of the quotation marks have been removed, as seen in the Track Changes version of the revised manuscript. Similarly, we have to acknowledge that words like ' verify' and 'expect' were too frequent in our text. In the revised version, they have in most places been replaced by other words or text.